# Homeostatic bidirectional plasticity in upbound and downbound micromodules in a model of the olivocerebellar loop

Elías M. Fernández Santoro[1], Lennart P.L. Landsmeer[1,2], Said Hamdioui[2], Christos Strydis[1,2], Chris I. De Zeeuw[1,3], Aleksandra Badura[1]*, Mario Negrello[1,4]*

**1** Department of Neuroscience, Erasmus MC, Rotterdam, The Netherlands, **2** Quantum & Computer Engineering Department, Delft University of Technology, Delft, The Netherlands, **3** Netherlands Institute for Neuroscience, Royal Academy of Arts and Sciences, Amsterdam, The Netherlands, **4** Department of Bionanoscience, Kavli Institute of Nanoscience Delft, Delft University of Technology, Delft, The Netherlands

\* m.negrello@erasmusmc.nl (MN); a.badura@erasmusmc.nl (AB)

## Abstract

Olivocerebellar learning is highly adaptable, unfolding over minutes to weeks depending on the task. However, the stabilizing mechanisms of the synaptic dynamics necessary for ongoing learning remain unclear. We constructed a model to examine plasticity dynamics under stochastic input and investigate the impact of inferior olive (IO) reverberations on Purkinje cell (PCs) activity and synaptic plasticity. We explored Upbound and Downbound cerebellar micromodules, which are organized loops of IO neurons, cerebellar nuclei neurons and microzones of PCs characterized by their unique molecular profiles and different levels of baseline firing. Our findings show synaptic weight convergence followed by stability of synaptic weights. In line with their relatively low and high intrinsic firing, we observed that Upbound and Downbound PCs have a propensity for potentiation and depression, respectively, with both PC types reaching stability at differential levels of overall strength of their parallel-fiber (PF) inputs. The oscillations and coupling of IO neurons participating in the Upbound and Downbound modules determine at which frequency band PFs can be stabilized optimally. Our results indicate that specific frequency components drive IO resonance and synchronicity, which, in turn, regulate temporal patterning across Upbound and Downbound zones, orchestrating their plasticity dynamics.

## Author summary

The olivocerebellar system is a part of the brain that facilitates learning and controlling movements. To coordinate movements it integrates sensorimotor information with motor command signals. The resulting behaviour needs to be continuously adjusted during motor learning. The leading hypothesis is that changes in strength of the synaptic connections between neurons underlie the learning

**Data availability statement:** The source code used to produce the results and analyses presented in this manuscript are available from GitHub repository: https://github.com/eliasmateo95/CerebellarLoop.

**Funding:** This work was supported by the Netherlands Organization for Scientific Research (NWO) VIDI/917.18.380,2018/ZonMw and the NWO NWA-ORC 2022 SCANNER (AB), the Erasmus MC Convergence Health and Technology Integrative Neuromedicine Flagship Program (AB and MN), the 2021 H2020 funding (n°650003) (MN). Financial support to CS provided by the European-Union Horizon Europe R&I program through projects SEPTON (no. 101094901) and SECURED (no. 101095717) and through the NWO - Gravitation Programme DBI2 (no. 024.005.022). Financial support to CIDZ was provided by the Netherlands Organization for Scientific Research (NWO-ALW 824.02.001), the Dutch Organization for Medical Sciences (ZonMW 91120067), Medical Neuro-Delta (MD 01092019-31082023), INTENSE LSH-NWO (TTW/00798883), ERC-adv (GA-294775) and ERC-POC (nrs. 737619 and 768914); The NIN Vriendenfonds for Albinism as well as the Dutch NWO Gravitation Program (DBI2). EMFS received salary from the 2021 H2020 funding (n°650003). LPLL received salary from the SECURED (no. 101095717) program. AB received a salary from the Netherlands Organization for Scientific Research (NWO) VIDI/917.18.380,2018/ZonMw grant. The funders had no role in study design, data collection and analysis, decision to publish, or preparation of the manuscript.

**Competing interests:** The authors have declared that no competing interests exist.

process. The challenge is to elucidate the factors that determine the exact timing and precision of the learned movements. To answer this question we developed a computational model of the two main types of modules of the olivocerebellar system that can control different types of movements in a bidirectional fashion. We found that the rhythm and coupling of the olivary neurons play an important role in controlling and stabilizing plasticity in the cerebellar cortex of both types of modules, together shaping learning-dependent timing of motor behaviour.

## Introduction

Procedural learning requires integration of sensory inputs with motor outputs, demanding circuits capable of both plasticity and stability. Sensorimotor control systems therefore rely on recurrent loops with feedback connections, enabling adaptive changes in synaptic strength based on ongoing activity patterns [1–3]. Among these systems, the olivocerebellar system stands out for its well-characterized plasticity mechanisms that are tightly linked to sensorimotor contingencies [4]. The olivocerebellar loop consists of the cerebellar cortex (CC), cerebellar nuclei (CN) and the inferior olive (IO), which feeds back into the CC via the climbing fibers (CFs) [5]. The sole output of the CC is generated by the inhibitory Purkinje cells (PCs), which receive sensorimotor information from the excitatory mossy fiber (MF) - parallel fiber (PF) pathway as well as the excitatory CF pathway.

The extensive dendritic trees of PCs receive signals from many PFs, all of which originate from granule cells [6,7]. Together with the inhibitory molecular layer interneurons (MLIs), the excitatory PFs can modulate the simple spike (SSpk) activity of PCs, allowing for precise bidirectional control [8,9]. Individual PFs form relatively weak synaptic connections with PCs, [10], but as a group they can readily double the intrinsic SSpk firing frequency when needed for online sensorimotor control [11]. In contrast, in most mammals, each PC receives a single CF from the IO, which delivers a powerful excitatory input resulting in an all-or-none complex spike [(CSpk); [4]]. Different from the SSpks (~50–100 Hz), the CSpks occur at a relatively low firing frequency (~1 Hz) during spontaneous activity and they can increase their instantaneous firing frequency about ten-fold during modulation [8]. Since the CSpks are triggered by the IO activity [12], they reflect the integration of its excitatory signals from one or more of the ascending or descending afferent systems [13,14] and its inhibitory input signals from the hindbrain [15–18]. Interestingly, the nuclei that form the sources of the excitatory inputs to the IO are largely overlapping with the sources of the MF pathways that provide the sensorimotor signals to the CC via the PFs [19].

During learning, PF to PC synapses undergo activity-dependent changes guided by the CF signals [20]. Classical models describe long-term depression (LTD) of these synapses when PF and CF are co-activated within a short time window [20,21], and long-term potentiation (LTP) when PF activity occurs outside this window [22]. However, alternative plasticity mechanisms have been proposed, where bidirectional and even inverse plasticity patterns have been observed depending on the pattern of

PF input, the intrinsic firing properties of PCs, and the local microcircuit context [23–27]. Maintaining a balance between potentiation and depression is crucial to avoid unbounded growth or elimination of synaptic weights [28–30], enabling the PF-PC synapses to remain responsive while supporting long-term adaptation to input statistics.

Question remains how stability can be achieved in the olivocerebellar loops with ongoing sensorimotor activity and following procedural learning. Several studies have shown that activity-dependent synaptic plasticity is generally rather unstable in recurrent circuits [1,31,32], often leading to difficulties with network dynamics [1,30,33–36]. Solving this question for the olivocerebellar system becomes particularly challenging when taking into account the heterogeneity of this network. Specifically, PCs in different cerebellar microzones display distinct intrinsic properties, with the PCs expressing Aldolase-C (also known as Zebrin II-positive PCs) typically exhibiting lower spontaneous SSpk firing frequency (30–90 Hz) compared to Aldolase-C-negative PCs (60–120 Hz) [9,37,38]. These electrophysiological differences are accompanied by divergent patterns of connectivity and plasticity: Aldolase-C-positive and -negative zones differ in their afferent and efferent projections [37,39–42], and they exhibit different propensities for long-term synaptic plasticity at their PF to PC synapses [43].

Building on these distinctions, the conceptual framework of so-called "Upbound" (Aldolase-C-positive) and "Downbound" (Aldolase-C-negative) modules was introduced to capture the observation that different microzones support distinct plasticity regimes during learning [44–46]. In particular, because of their relatively low baseline firing frequency, the Upbound modules are characterized by a propensity for LTP over the course of training. Conversely the Downbound modules, which show high levels of baseline activity, show a propensity for LTD [9,43,47–49].

Our model explores the existence of common mechanisms in the olivocerebellar system that can stabilize changes in synaptic weights across modules that support distinct behaviours. We address this question by modeling how stability can emerge in a system where plasticity is continuously shaped by the circuit's own reverberating activity. Unlike recurrent excitatory networks, where Hebbian plasticity often leads to runaway instability, the olivocerebellar system forms a closed loop with distinct architecture: feedforward excitation, delayed inhibitory feedback, and intrinsic STOs in the IO. While this system lacks recurrent excitation, it still poses challenges for stability as the PF-PC synaptic plasticity is coupled to internally generated output. In this context, stability does not mean averting unbounded weight growth, but rather the emergence of robust synaptic weight distributions under ongoing plasticity and stationary input statistics. These dynamics reflect a homeostatic baseline regime in which Upbound and Downbound (Up/Downbound) modules emerge as stable configurations, rather than as pre-imposed initial conditions.

We hypothesize that the resonant dynamics of the olivocerebellar loops, particularly those shaped by the IO STOs and inhibitory feedback, are key to stabilizing PF-PC synaptic weights during procedural learning in both Up/Downbound modules. These dynamics may serve to temporally align CF input with relevant PF activity patterns, thereby promoting structural, frequency-dependent synaptic modifications across repeated exposure to the similar input temporal regularities. In other words, establishing robust encoding of learned associations between movements and the predictions of their sensory outcomes [44].

To test this hypothesis, we developed a biologically grounded network model of the olivocerebellar system. The model incorporates reverberating, quasi-periodic activity [50–52] and simulates micromodules with distinct electrophysiological properties of PCs, reflecting Up/Downbound zones. A bidirectional, homeostatic plasticity mechanism, a Bienenstock-Cooper-Munro-type (BCM) rule, jointly with an LTD vs. LTP mechanism, governs learning at the PF-PC synapse. These rules are not used as a predefined stability constraint, but rather to investigate how internally generated activity can lead to stable and interpretable synaptic configurations under learning-like conditions. In our model, synaptic weights represent the efficacy of the PF input onto the PCs and evolve over time according to the BCM rule. These weights modulate the postsynaptic current (PSC) generated by each PF bundle, which in turn shapes the SSpk activity. The model includes the core components of the olivocerebellar loop (PCs, CN and IO neurons) with distinct parameter regimes corresponding to Up/Downbound modules.

While the model simulates synaptic plasticity and evolving weight dynamics, it does not incorporate explicit motor error signals or task-dependent modulation of CF activity. Instead, it focuses on exploring how internally generated loop dynamics organize synaptic weights in the absence of explicit motor error feedback. As such, the model captures baseline synaptic homeostasis under fixed, ongoing input conditions rather than procedural learning per se. In this context, it examines how the olivocerebellar loop can autonomously stabilize its synaptic configuration, shaped by the interaction of intrinsic excitability, PF drive, and IO feedback. We propose that these stable states may serve as a scaffold upon which learning-related, supervised plasticity mechanisms (driven by behaviorally relevant CF signals) can subsequently act. To test whether the stable Up/Downbound baseline states generated by our model could support learning-related plasticity, we simulated a classical eyeblink conditioning paradigm. Our data show that the stable Up/Downbound states persist and, upon stimulus pairings, undergo distinct plasticity-driven changes in firing, consistent with experimental findings.

Our simulations show that the differences between Up/Downbound zones are not solely due to intrinsic properties of their PCs, rather, they also emerge from the dynamics of the olivocerebellar loop. Moreover, different PF input frequency bands differentially drive CSpk responses, suggesting that specific frequency components promote IO resonance and this frequency selectivity is further shaped by IO gap junction coupling. Together, our data highlight that temporal pattern selectivity processing across both Up/Downbound micromodules is enhanced and stabilized by reverberations in the olivocerebellar loop during procedural learning.

## Results

### Designing an olivocerebellar loop model to examine plasticity dynamics

Our hypothesis is that PF-PC synaptic weights are stabilized during procedural learning by the resonant dynamics of the olivocerebellar loops. To test this, we developed a biologically plausible model of the olivocerebellar system (**Fig 1A**), which is decoupled from the rest of the brain to focus on the olivocerebellar resonant network dynamics that drive plasticity. The questions we tested with this model are centered on exploration of synaptic stability during ongoing plasticity. We emulated ongoing arbitrary cortical and subcortical inputs to the cerebellum via 5 uncorrelated PF bundles, which is conveyed by 100 PCs via 40 CN cells to 40 cells in the IO [53–60]. These IO cells have the ability to respond by advancing or delaying their phase as a function of when inhibition arrives in the oscillation cycle, as observed in *in vivo* experiments [16,61–65].

As input to our model network we used bundles of PFs as a noisy current input rather than individual synapses. A bundle of PFs represents the combination of numerous synaptic sources, as expected during the awake and behaving state [66,67]. To avoid biasing the input, we modelled background activity as the sum of several independently generated Ornstein-Uhlenbeck (OU) processes, taken to represent the integrated, continuous current from large numbers of PF EPSCs [68]. This use of stochastic but temporally correlated input allowed us to assess whether synaptic weights stabilize under a fixed input structure, enabling us to isolate the contribution of internal loop dynamics, such as IO resonance and feedback, to plasticity. This choice was motivated by theoretical considerations and in vivo experimental data of granule cell and PF activity during learning, at rest and during walking [66,69,70], which shows complex, high dimensional firing patterns transmitted to PCs in awake animals. We refer to this temporally varied OU input as "frozen input" because it was generated once and then held constant across all simulations. The input was not repeated or manipulated across trials; it remained fixed throughout all simulations, serving as a controlled condition. This design allows us to compare network behavior across simulation conditions while ensuring that any observed differences could be attributed exclusively to the effects of plasticity mechanisms, rather than to variations in the external input. This is a crucial aspect of our model as we can test the hypothesis that under the stimulus representing sensorimotor contingencies there would be synaptic adjustment and weight stabilization in the PF bundles.Our PF bundles can be seen as bundles of correlated input originating via MF rosettes [10,71]. Given that around 90% of PF synapses are silent [72], we modelled 5 bundles with approximately 200 active PFs each (Fig 1A–C). The PF-PC synapse acts as a "meta-synapse", representing the average direction of

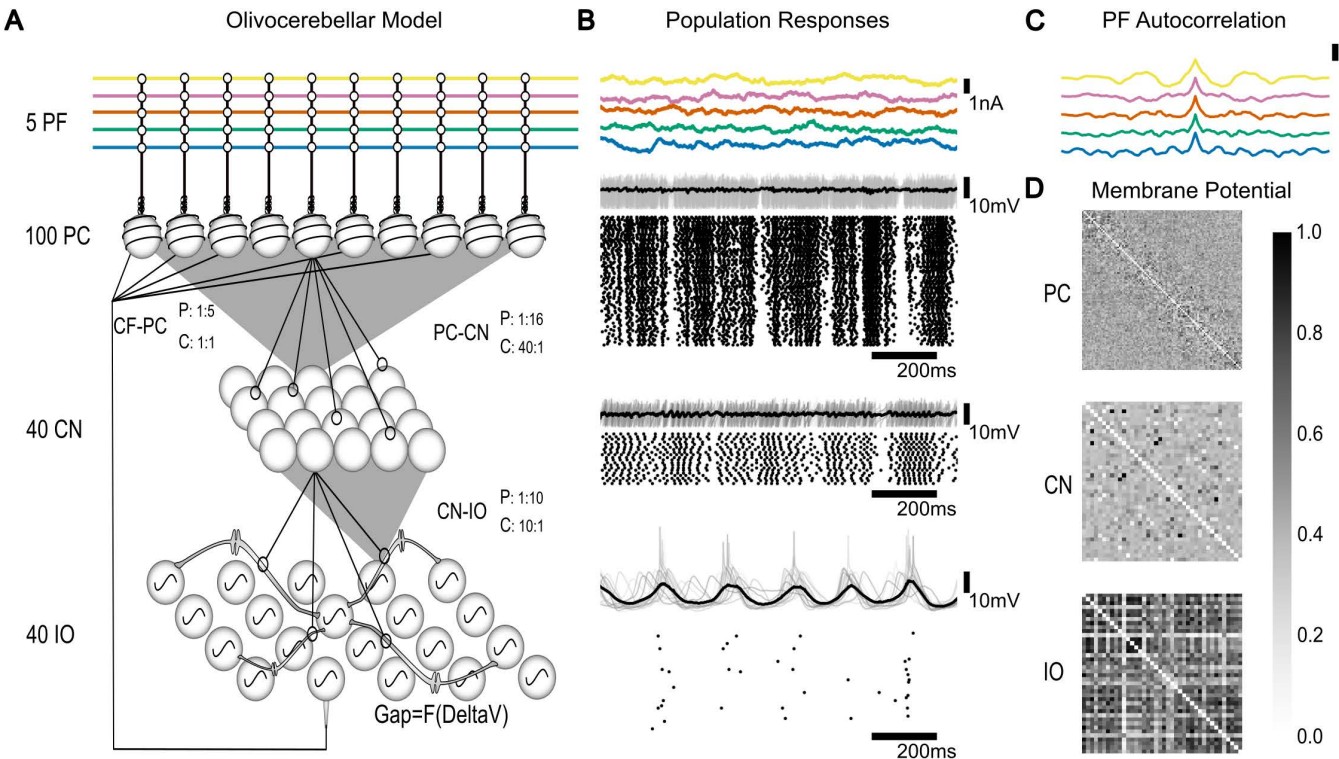

**Fig 1. The olivocerebellar loop model and its population response (Downbound, coupled). (A)** Diagram of the olivocerebellar loop model. It consists of 5 PF bundles, 100 PCs, 40 CN neurons and 40 IO neurons. The projection (P) and convergence (C) ratios between neuronal populations are indicated for each connection between populations. Specifically, our model has 5 PF bundle inputs into 100 PCs. Each PC projects to 16 CNs. The CN layer consists of 40 CN neurons, which receive input on average from 40 PCs (from 30 to 52 PCs). Each CN projects to 10 IO cells. There are 40 IO cells, each of them receiving on average 10 CNs (from 6 to 16 CNs). The IO cells are modelled with gap junctions and subthreshold oscillations. Half of the IO cells project back to the PCs with each of those IO cells projecting to on average 5 PCs (from 2 to 9 cells), while each PC only receives signals from 1 IO. Each IO neuron connects to all other IO neurons in the population via the gap junctions. **(B)** From top to bottom: the PF bundle input modeled as Ornstein-Uhlenbeck processes; PC responses, CN responses and IO responses (membrane potential and raster). **(C)** Autocorrelation of each PF input. **(D)** Heatmap of the correlation of membrane potentials within each population.

change of the individual synaptic weights in the PF bundle. To represent plasticity, we added a weight value for each PF bundle, which could change independently. Model parameters were tuned to match spontaneous activity statistics reported experimentally — such as PC SSpk and CSpks firing rates, CN output, and IO STOs. No tuning was performed to enforce the hypothesized plasticity directions. Rather, the observed phenomena (e.g., synaptic weight dynamics, CF pause modulation, and frequency-dependent IO entrainment) emerged from the interactions of BCM and LTD plasticity, IO coupling, and intrinsic PC excitability under fixed input conditions.

We tuned the PCs firing frequency to represent a mixture of their intrinsic firing frequency and their response to the PF input. As shown experimentally [9], when the PF input was completely absent, PCs had an intrinsic firing frequency that was about 20 Hz lower than when the PF input was present. For simplicity we focussed on the primary drivers of the olivo-cerebellar loop dynamics - the GABAergic CN projection cells, and excluded cells involved in other loops. CN AdEx model cells were primarily designed to have robust transmission of rate coding [73]. Our CN cells received divergent input from the PC layer, which modulated CN responses in an anti-correlated fashion [73,74].

It is known that the IO contains a mixture of cells: some of which oscillate robustly, some display dampened oscillations, and others that do not seem to display any resonant behavior. Among the oscillating cells, many exhibit a low-frequency

rhythm in the 4–10 Hz range [52]. Our modelled IO cells oscillated robustly around 5.8-6 Hz and were connected with electrotonic Connexin-36 synapses [52,75]. IO spiking synchrony was disrupted in absence of gap junction coupling (S1 Fig). Though anatomical clusterization of cells is highly plausible [55,76], here we examined a small set of cells without explicit clustering.

## Homeostatic plasticity and CSpk-triggered plasticity mechanisms

To capture the essential interactions between LTP and LTD we modelled the PF-PC synapse with a mixture of a BCM learning rule and a PF input dependent CSpk-triggered LTD mechanism. This CF-induced LTD operates inversely to the classical BCM mechanism and changes the probability of BCM-LTP [12]. In our model, the synaptic weights were potentiated when PC activity was low to maintain excitability. This complex combination of synaptic plasticity mechanisms enabled us to inspect their interactions and dynamics.

The BCM mechanism (**Fig 2A**) in our model encapsulates the effect of changes in the PF activity by either potentiating or depressing the synapse independently of CF input. This mechanism depends on 3 factors: the average of the recent activity of the PC, $\rho_{PC}$; the BCM threshold $\theta_M$ — which also depends on $\rho_{PC}$— and the instantaneous activity of the PF, $\rho_{PF}$ (**Fig 2B**). After an IO spike (**Fig 2C**), a pause in SSpks leads to a CF pause, which causes a decrease in $\rho_{PC}$. The LTD component of plasticity due to the CSpk gives a negative change in synaptic weight also proportional to the $\rho_{PF}$ (**Fig 2D**). The plasticity threshold $\theta_M$ distinguishing depression from potentiation follows $\rho_{PC}$, that is $\theta_M$ lower than $\rho_{PC}$ results in potentiation. Conversely, $\theta_M$ higher than $\rho_{PC}$ leads to depression. Both these changes are proportional to activity $\rho_{PF}$ for each synapse (see **Fig 2E**). The total change in synaptic weight is computed as the sum of two plasticity components: a BCM-based continuous update and a CSpk-triggered term (**Fig 2F**). The BCM plasticity mechanism has only one hard constraint limiting the potentiation of the PC to frequencies of 250 Hz, as firing rate beyond this rate is considered pathological.

After an initial transient, synaptic weights converged to stable distributions. However, we observed small residual stabilizing oscillations around this stable mean, which we refer to as weight fluctuations. These reflect the continuous balancing process of BCM potentiation and CSpk-triggered LTD under stochastic input. Importantly, these fluctuations were minute relative to the absolute weight values and did not show cumulative drift. While shaped by the structure of the input —such as frequency content and temporal correlations— the fluctuations were not dependent on the input duration, indicating a steady-state regime rather than input-length dependence.

While the BCM rule used is not fitted to *in vitro* data, it captures key features of activity-dependent potentiation and depression shaped by CSpk activity. This allowed us to examine how loop-level dynamics guide synaptic trajectories. Different BCM parameter choices may alter the speed of convergence but not the main results: weights stabilized, CSpk frequency remained frequency-dependent, and Up/Downbound modules retained distinct profiles. For example, we selected the threshold time constant ($\tau_M$) within a range of 10–20 ms, based on the typical duration of CF pauses. This parameter governs the low-pass filtering of recent PC activity used to compute the threshold $\theta_M$. We did not tune $\tau_M$ to produce specific outcomes; rather, this range was chosen *a priori* based on known physiology, and only *a posteriori* did we find that it yielded expected plasticity directions —net depression in Downbound zones and potentiation in Upbound zones.

## Synaptic weights are stable across plasticity epochs

We hypothesized that synaptic weights would reach a new level of stability and stay at the level for subsequent plasticity events. To investigate these changes in synaptic weights across plasticity events we did the following simulations (see Methods), where for each epoch we gave the same frozen input as PF input. We started with the first epoch ("no plasticity") with the plasticity turned off and with static synaptic weights which were constant throughout this 120 s experiment. After this "no plasticity" epoch finished, BCM and LTD plasticity mechanisms were enabled and the same frozen input was replayed. At the end of this plasticity epoch, we took the last value of the filtered synaptic weights and turned it into the

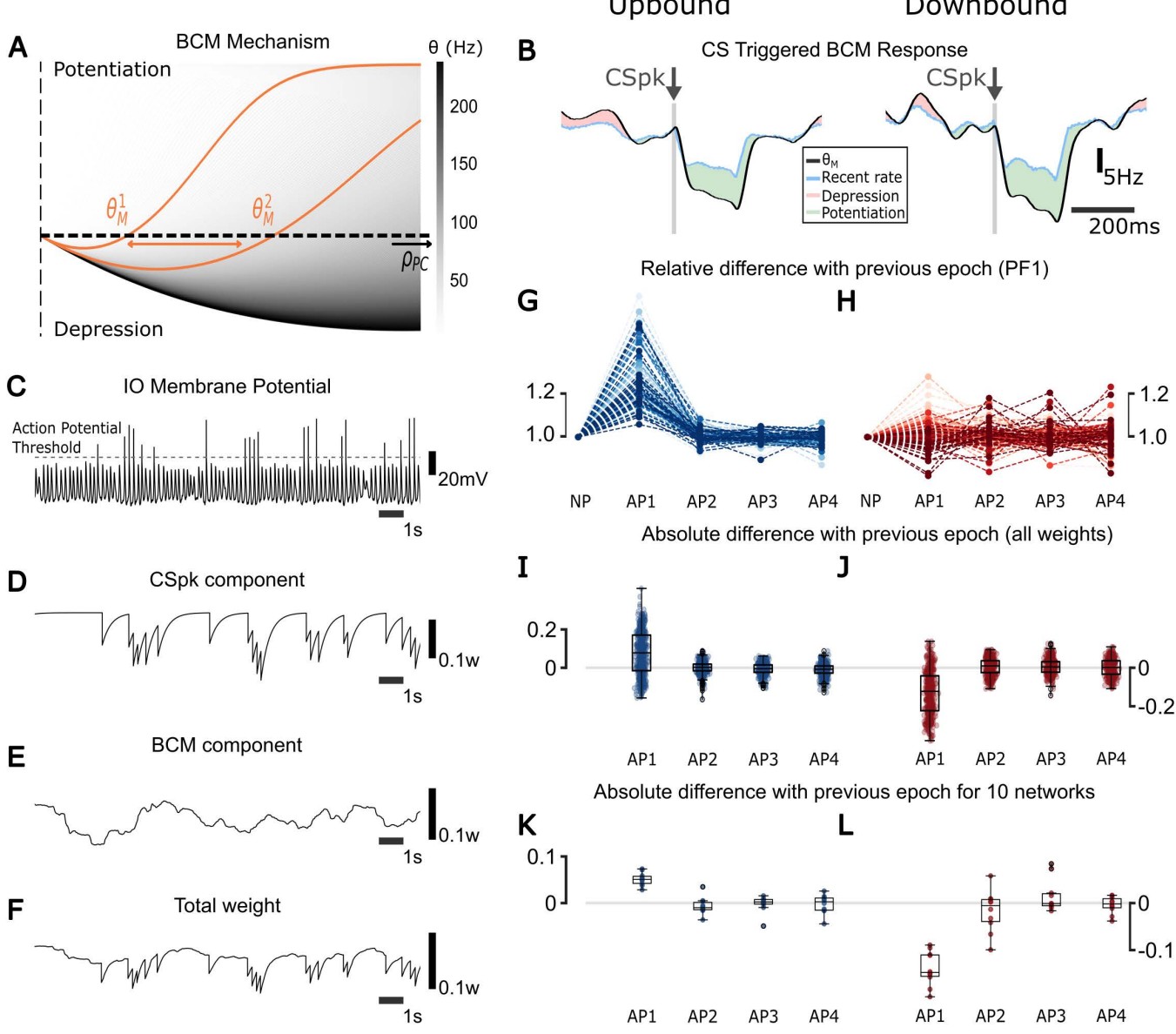

**Fig 2. The plasticity direction differs between Up/Downbound micromodules. (A)** BCM mechanism: the x-axis represents the activity of the PC, and the y-axis represents the predicted plasticity change at a PF-PC synapse before scaling by PF activity. Positive values indicate potentiation; negative values indicate depression. Each gray curve shows the BCM function for a different value of the plasticity threshold θ. *The colormap encodes the corresponding θ values (in Hz), with lighter gray indicating lower thresholds. As PC activity increases, the sliding threshold θ shifts rightward, reducing the potentiation range and increasing the chances of depression (a larger part of the curve is negative). The same applies for a lower activity in PC, leading to higher chance of potentiation.* **(B)** CSpk-triggered change in BCM. When there is a CSpk, the activity of the PC (blue) decreases. The sliding threshold θ *(black) follows and goes lower than the PC activity, leading to potentiation (green). When θ is higher than the activity there is depression (red). This is shown for both Up/Downbound zones.* **(C)** Trace of the IO membrane potential for Downbound coupled scenario. This trace was selected to illustrate the short-timescale effects of different IO spike intervals on plasticity mechanisms (BCM and CSpk-triggered LTD). While this particular trace appears bursty, it is not representative of the full IO population. IO neurons in the model exhibit a range of firing modes including tonic firing and quiescence. The mean response profile shows a ∼ 1 Hz firing rate (**Fig 1B**) and the IO has a large distribution of firing rates across the population (**Fig 3B**). IO burstiness was not explicitly tuned and remained within biologically observed ranges (<6Hz), consistent with experimental findings [52,77–80]. **(D)** CSpk-triggered LTD component of synaptic plasticity. This shows the change in synaptic weight due to a IO spike (CSpk) event, proportional to the instantaneous activity of the PF. In the absence of a CSpk, this component is zero; following a CSpk, it produces a transient depression that decays back to zero. **(E)** BCM component of synaptic plasticity. This reflects the change in synaptic weight based on the recent activity of the PC and the BCM threshold θ. *It evolves continuously, independently of CF input.* **(F)** *Total synaptic weight update, computed as the sum of the BCM component and the*

*CSpk-triggered LTD component. All values are unitless and represent normalized synaptic efficacy changes per time step in the model.* **(G)** *Relative difference of synaptic weights of all PCs connecting to PF 1 before and after each plasticity (AP 1, 2, 3 and 4) between the epochs (current weight divided by the one of the previous event) for Upbound zones.* **(H)** *Same as I but for the Downbound zones.* **(I)** *Boxplots depict absolute differences between the epochs for the whole population of weights (current weight minus previous weight) for Upbound zones.* **(J)** *Same as K but for the Downbound zones.* **(K)** *Median synaptic weight changes across four plasticity epochs for 10 independent Upbound simulations with randomized network and OU input per run.* **(L)** *Same as M, but for Downbound simulations. Synaptic weights are unitless and represent normalized efficacy values in the model.*

new static weights for the final epoch (after plasticity). Again the same input was given one last time with the new static weights. This configuration enabled us to compare synaptic weights at the PF-PC synapse before and after plasticity. To evaluate the impact of plasticity on the weights, each experiment was performed 4 times with the same frozen input (Fig 2G-L).

The same frozen input was used in each epoch. It provides consistent, varying input statistics and temporal correlations across epochs ensuring reproducibility. When plasticity is switched off, the synaptic weights remain fixed, allowing us to compare resonant activity at stable weight configurations before and after plasticity. In contrast, during plasticity-on epochs, the BCM and CSpk-triggered LTD mechanisms allowed the weights to settle into a new homeostatic equilibrium.

For both Up/Downbound zones the direction of change of the 500 synaptic weights for the 100 PC population (5 synapses per PC) was defined during the first "plasticity" epoch. The synaptic weights at this epoch were reduced in the Downbound zone and increased in the Upbound zone (Fig 2I and 2J), suggesting that the intrinsic PC currents determined the plasticity direction. From the second epoch after plasticity we found that the weights were mostly stable, though a few changed in rank (Fig 2G and 2H), indicating that the interaction between CSpk and PF inputs may drive these rank changes. In Downbound zones weights fluctuated around the mean, indicating a faster rate of weight change, due to the higher intrinsic PC current (Fig 2J). The Upbound zone, with a lower intrinsic drive, exhibited more synaptic weight stability after relaxation in epochs 3 and 4 (Fig 2G and 2H).

To test the robustness of these weight dynamics, we repeated the simulation protocol using 10 different random seeds, each with independently generated OU input and randomly sampled network parameters. For each seed, we computed the median weight change per plasticity epoch (Fig 2K and 2L). The same bidirectional trend was observed across all simulations: Upbound modules exhibited early potentiation followed by stabilization, while Downbound modules showed early depression followed by stabilization. These results support the generalizability of our findings and indicate that the observed stabilization arises from the intrinsic dynamics of the model rather than from input-specific contingencies.

## Gap junction coupling creates a global phase and leads to synchronized firing

We expected gap junctions (i.e., "coupling") to have an impact on IO STO synchrony with strong coupling leading to higher synchrony. To quantify this effect, we looked at the STO oscillation phases over time. These were decomposed into a network level (global) phase, defined as the mean Hilbert phase across all IO neurons, and into an individual-level (delta) phase, defined as the difference between the Hilbert phase of each IO neuron and the global phase at each time point. This was done to compare the difference between the individual and global phases (S2A Fig). The global phase of STOs, representing average phase progression, was very stable in both coupled and uncoupled conditions (S2B, S3B Fig). In the condition with coupled IO cells, oscillators phase-locked onto the global phase creating synchrony (S2C Fig) not seen in the uncoupled case (S3C Fig), where oscillators behaved independently. To investigate whether the coupling also led to dynamically organized clusters in the IO, we calculated the pairwise Kuramoto phase (S2D Fig). At the scale of one second (S2E Fig), transient clustering could be observed, however clusters were not stable (S2F–G Fig) and reflected the random phase fluctuations driven by GABAergic CN input. At the scale of 30 s (quarter-time of the full simulation) no clear clustering was observed. Network synchrony peaked around IO spikes, but only for the coupled scenario (S3A Fig).

## Upbound PCs have a propensity for potentiation and Downbound PCs for depression

As it has been shown that PCs adaptively adjust their SSpk firing rate via CSpks to elicit blinks during eyeblink conditioning [45,81,82], we hypothesize that the direction of change of the PC firing rate depends on the intrinsic frequency of each PC microzone. Eyeblink conditioning is a classical associative learning paradigm where a neutral stimulus, such as light, is paired with an air puff that evokes reflexive eyelid closure so that the initially neutral stimulus can eventually trigger blinking. As argued previously, in this associative task, bidirectional changes in the SSpk activity are expressed in the Downbound zone during initial stages of training, whereas the Upbound zone controls acceleration of movement in paradigms such as the vestibulo-ocular learning (Fig 3A) [44]. In fact, several studies support the distinction between these zones, proposing that they constitute complementary modules that may act in concert to encode prediction and feedback signals during sensorimotor learning [5,9,43,47–49]. Notably, while eyeblink conditioning involves learning over many trials with discrete, stimulus-locked inputs and explicit error signals, our model instead uses fixed, temporally varied, OU input, which was generated once and reused across simulations to explore whether intrinsic firing properties and internal dynamics alone can explain the directionality of plasticity. Fig 3A provides a schematic summary of previously published experimental findings [44], illustrating the typical direction of SSpk modulation in Up/Downbound PCs during eyeblink conditioning and vestibulo-ocular learning. In contrast, Fig 3B shows how the same directionality emerges in our model when

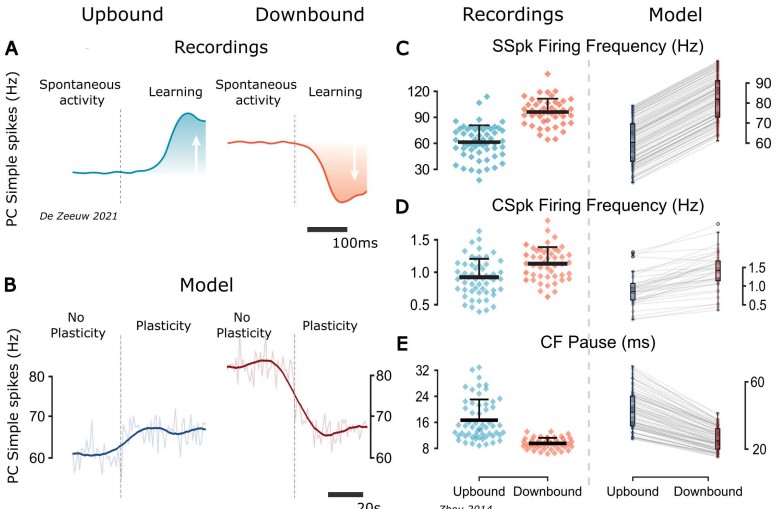

**Fig 3. Comparison of experimental results and model results for Downbound and Upbound zones. (A)** *Schematic summary (adapted from [44]) of SSpk modulation direction in Up/Downbound modules during eyeblink conditioning and vestibulo-ocular learning. This is not a direct experimental trace but a schematic summary of findings based on in vivo PC recordings during eyeblink conditioning (Downbound) and vestibulo-ocular learning (Upbound).* **(B)** *Model results showing population-averaged PC SSpk activity across the three epochs (no plasticity, plasticity, after plasticity). Thin translucent trace: raw 1-s binned population mean across PCs. Thick trace: Savitzky-Golay-smoothed version of the same trace (window length 110 s, polynomial order 9), applied across the entire concatenated time series. Minor boundary effects occur at the epoch transitions (vertical dashed lines). Note that the timescales differ: in the model changes emerge over seconds, whereas in experiments they appear over many trials and are expressed in relation to stimulus onset within hundreds of milliseconds.* **(C-E)** *Comparison of experimental extracellular recordings in awake mice, at-rest, from Up/Downbound-identified PCs, adapted from [9] and model results of PC firing frequency, CSpk firing frequency and CF pause duration. Each gray line connects to the paired value for a single model cell across the two modules. These pairs share all parameters except for: (1) the intrinsic firing rate of the PCs and (2) the $g_{Cal}$ and $I^{(IO)}_{OU}$ parameters of the IOs, as shown in Table 7. There is a significant difference in means across Up/Downbound modules for the SSpk (**t=-13.78, p<0.001; two-sample Student's t-test; n=100**), CSpk (**t=-4.59, p<0.001; two-sample Student's t-test; n=40**), and CF pause (**t=-12.52, p<0.001; two-sample Student's t-test; n=40**)*, *which also has a smaller variance in the Downbound zone (**F=2.94, p<0.001; F-test using CCDF of F-distribution; n=40**). Note that the model traces reflect activity during the plasticity epoch, before full stabilization, and are used to assess directionality of plasticity rather than absolute firing rates. Moreover, "before plasticity" refers to the initial model state prior to synaptic adaptation and while it represents a biologically plausible starting point, it does not reflect a stable physiological state. The "after plasticity" condition corresponds to the stationary state of the network under ongoing synaptic adaptation, and should be used as the model's functional baseline.*

plasticity mechanisms (BCM, LTD, LTP) are activated, but over much longer timescales (seconds rather than hundreds of milliseconds. For display, we plot the raw 1-s binned population mean together with a Savitzky-Golay-smoothed trace. The smoothing applied across the full concatenated simulation produces small boundary effects at the epoch transition. Nevertheless, the comparison highlights that, despite the different inputs and timescales, the baseline level of PC activity is sufficient for the BCM mechanism to establish the direction of synaptic weight change. PCs in our model were tuned to reproduce the experimental data (Fig 3C and 3D), whereby Downbound PCs showed relatively high SSpks (~90 Hz) and CSpks (~1.3 Hz) firing frequencies, while Upbound PCs had lower average SSpk (~60 Hz) and CSpk [(~0.9 Hz), [9,44]]. Tuning PCs intrinsic currents resulted in the CF pause reproducing experimental observations (Fig 3E).

The direction of plasticity was the same irrespectively of the absence or presence of IO gap junction coupling (S4A-E Fig). For the Downbound micromodule, there was a small increase in the mean of the CSpk firing frequency in the uncoupled case with respect to the coupled case (Figs 3D and S4B), which led to a wider distribution of the CSpk frequency squared coefficient of variation [(CV2); (S4F Fig)].

### Inhibition followed by disinhibition of CN elicits CSpk

We hypothesized that specific characteristics of the input would elicit CSpks. We started by analysing the CSpk-triggered PSC (Fig 4A) and we observed that the CSpks were promoted by specific oscillations present in the input. In our model, the PSC refers to the current that enters the PC soma,specifically, the sum of the 5 PF currents, each scaled by their corresponding synaptic weight. CSpks were evoked when the input carried specific phases and frequency (around 5–6 Hz), with a preference for reduced input just before the CSpk. At the level of the CN this translated to inhibition followed by disinhibition, which had previously been empirically observed by [83,84]. The decrease (following an increase) in PSC happened 60 ms before the bottom of the subthreshold oscillation (STO) of the IO membrane response, which was closely related to the synaptic delay in the loop (10 ms between PC and CN and 50 ms between CN and IO). This suggests that CSpks had a preference for particular frequencies and phases of the input. A clear contrast is observed after plasticity, as in Upbound zones the sensitivity to the PSC oscillations was increased and reduced for Downbound zones.

Every CN spike contributed to a reset of the IO STOs (Fig 4B), in line with experimental results [16]. We found that in the Upbound zone there was a lower CN spike frequency and lower amplitude of IO oscillations before the CN spike. As there was more CN inhibition in the Downbound zone, a lower CN spike frequency led to reduced dampening of the IO oscillation. In addition, we found that increasing CN inhibition lowered the frequency of IO STOs while increasing IO spike (CSpk) output (S5E Fig). In the Upbound zone, we observed a lower CN spike frequency and lower amplitude of IO oscillations before the CN spike, indicating weaker modulation of IO dynamics. In contrast, in the Downbound zone, stronger CN inhibition slowed the IO STOs and reduced their amplitude, creating conditions that increased the likelihood of IO spiking. This may be due to reduced oscillation amplitude, resulting in milder hyperpolarization and thus a greater chance of crossing spike threshold. Thus, CN activity modulates IO excitability indirectly, through effects on STO phase and amplitude, rather than through direct excitation. These nonlinear interactions were further shaped by PF-PC plasticity: IO responses to CN spikes varied depending on prior synaptic changes, showing that the same input could produce variable IO responses depending on network history. This is consistent with experimental observations of trial-to-trial variability in CSpk timing [7,85], and supports the idea that IO STO phase and amplitude influence CSpk generation [52]. After plasticity, CN spikes no longer significantly affected IO STO frequency in the Downbound zone. This result suggests that the IO STOs are affected but not driven by CN inhibition.

After the first "plasticity" epoch, the PC SSpk activity was increased for the Upbound zone and decreased for Downbound zone, resulting in both PC populations firing at about 60 Hz after plasticity (Fig 4C). As a result, the CN activity was decreased for the Upbound and increased for the Downbound zone (Fig 4D). Furthermore, the CSpk frequency was decreased for Upbound PCs and increased for Downbound PCs (Fig 4E). The length of the CF pause decreased for both Up/Downbound zones (Fig 4F).

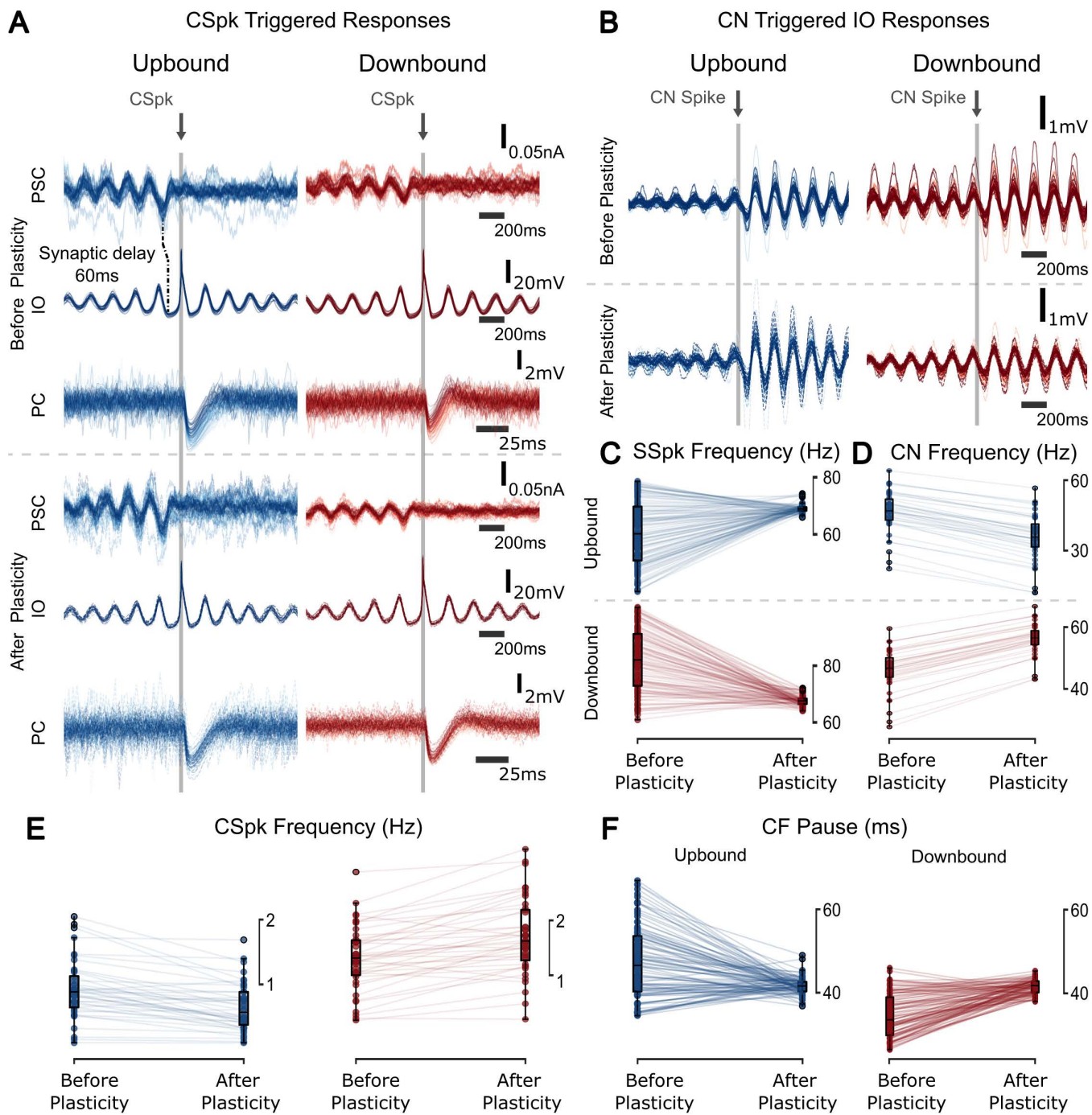

**Fig 4. Zone-dependent spiking resonances before and after plasticity. (A)** CSpk-triggered Postsynaptic current (PSC) recorded at the soma of the PC (top), IO membrane potential (middle), and the PC membrane potential showing the CF pause (bottom). Traces are aligned to the CSpk time (t = 0), but the time window range of the CF pause trace differs slightly to best illustrate its response. The synaptic delay between the PSC entering the PC, going through the CN (10 ms) and reaching the IO (50 ms). As the inhibitory CN input coincides with the bottom of the IO STO, a CSpk is elicited. **(B)** CN spike triggered membrane response of the IO, before and after plasticity. **(C)** Boxplots of SSpk frequency before and after plasticity. There there is a significant difference in the across the Up/Downbound zones before and after plasticity (before plasticity: **t = -13.90, p < 0.001,** after plasticity: **t = -5.36, p < 0.001; two-sample Student's t-test; n = 100**) and the variance is also different after plasticity across the zones (**F = 1.69, p = 0.005; F-test using CCDF of F-distribution; n = 100**). There is a significant difference across plasticity epochs for both zones (Upbound: **t = -7.64, p < 0.001,** Downbound: **t = 12.89, p < 0.001; two-sample Student's t-test; n = 100**). The variance of the SSpks after plasticity is also significantly different (Upbound: **F = 59.88,**

**p < 0.001**, Downbound: **F = 36.00, p < 0.001; F-test using CCDF of F-distribution; n = 100**). **(D)** Boxplots of CN spiking frequency before and after plasticity (before plasticity: **t = -3.26, p = 0.002**, after plasticity: **t = 9.15, p < 0.001**, Upbound: **t = 5.05, p < 0.001**, Downbound: **t = -7.26, p < 0.001**; **two-sample Student's t-test; n = 40**). **(E)** CSpk frequency before and after plasticity (before plasticity: **t = 3.98, p < 0.001**, after plasticity: **t = 9.44, p < 0.001**, Upbound: **t = 2.99, p = 0.004**, Downbound: **t = -2.81, p = 0.006; two-sample Student's t-test; n = 40**). The variance of CSpks is also different after plasticity across the zones (**F = 2.91, p < 0.001; F-test using CCDF of F-distribution; n = 40**). **(F)** CF pause length is significantly different before plasticity (**t = -12.52, p < 0.001; two-sample Student's t-test; n = 40**). The length of the CF pause is significantly modulated by plasticity in both zones (Upbound: **t = 6.37, p < 0.001**, Downbound: **t = -13.00, p < 0.001; two-sample Student's t-test; n = 40**). The CF pause variance was smaller after plasticity (Upbound: **F = 20.76, p < 0.001**, Downbound: **F = 9.34, p < 0.001; F-test using CCDF of F-distribution; n = 40**).

Investigating the effect of gap junction coupling, we observed that the uncoupled Downbound IO had on average a higher variance and higher CSpk firing frequency (S5B Fig), leading to a higher variance of SSpk firing frequency for Downbound PCs (S5A Fig). Nevertheless, the CN (S5C Fig) and CF pause (S5D Fig) were the same for both coupled and uncoupled conditions.

### Specific input frequency bands differentially drive plasticity

Our previous hypothesis was that specific characteristics of the input would directly influence CSpk generation. Here, we expand our focus to include the full olivocerebellar loop, proposing that distinct input frequency bands engage different resonant dynamics of the circuit, thereby shaping both CSpk timing and the resulting plasticity. As a continuation of the results shown in Fig 4, we examined how different frequency bands shape loop-level responses under plasticity. Signals from different sensorimotor sources have different spectral properties, and PC population SSpks activity has been reported to contain bands from low delta [86] to high gamma [50] frequencies.

To investigate how these spectral differences influence the loop dynamics under plasticity we filtered the original PF signal into distinct frequency bands. The filtered signals were normalized to keep the same spectral power (Fig 5A), and were used in exact copies of the same network to compare the before and after plasticity activity of the cells across the different frequency bands (Fig 5B-D). We found that CN dynamics were hardly modulated by the filter classes, indicating high level of signal divergence and mixing (Fig 5C).

As judged from both SSpk (Fig 5B) and CSpk (Fig 5D) distributions, Up/Downbound PC zones showed marked differences in how their firing rates responded to different PF input frequency bands. These differences were most pronounced for SSpks in the upbound zones.

While the direction of plasticity of CSpks activity qualitatively resembled physiological observations [9,43,44], frequency distributions were sensitive to plasticity under different bands, with lower input frequencies driving most of the variability. This was true for low frequencies (5–10 Hz band), where the CSpks activity was higher for Up/Downbound zones. Furthermore, frequency bands larger than 10 Hz evoked similar SSpks and CSspks distributions after plasticity, indicating that the PC network response was robust to frequency variations.

### Differential CSpk response to PF input changes drives plasticity in 5–10 Hz band

The largest differential plasticity responses were observed for the 5–10 Hz filtered input in the Downbound zone. Here we observed a large decrease in SSpk frequency after plasticity (Fig 5B), with a twofold increase of CSpks (S6A Fig). Before plasticity, the SSpk frequency was approximately the same for the 5–10 Hz filtered input and the "no filter" case, as the current input power spectrum and weights were the same. We assumed that the decrease in SSpk frequency after plasticity originated from CSpk-triggered LTD. However, this assumption did not explain the increase in CSpk frequency for the 5–10 Hz filtered PF input.

We looked at the relationship between the STO frequency and CSpk frequency (S6B Fig). We found that STO frequency and frequency-distribution width was narrower in the Downbound than in the Upbound zone. For both Up/

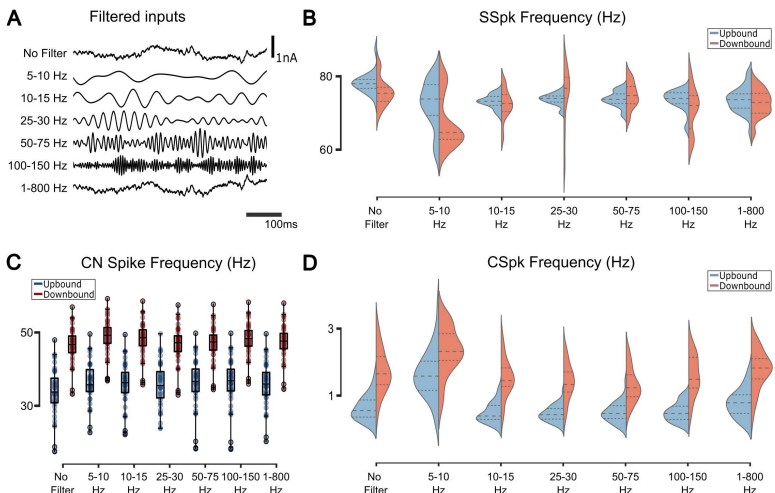

**Fig 5. Response of the olivocerebellar model after plasticity for Up/Downbound zones for filtered inputs. (A)** Different types of filters used on the original input (no filter). **(B)** When comparing with the no filter input, the distribution of frequency of SSpks is always different across frequency bands for the Upbound zone and in the case of the Downbound it is different between the 25-30 and 50-75 Hz frequency bands (for statistics see S1 Table). The distribution of the SSpk frequency is significantly different between Up/Downbound across the different filter inputs (see S1 Table for statistics). **(C)** The CN spike frequency stays the same for all types of inputs across modules (see S2 Table for statistics). **(D)** CSpks frequency stays the same for all types of inputs across modules (for statistics see S2 Table). The response of the IO varies significantly for the 5-10 Hz frequency band (Upbound: **t=-7.86, p<0.001** and **D=0.75, p<0.001**, Downbound: **t=-4.04, p<0.001; two-sample Student's t-test** and **D=0.5, p<0.001; two-sample ks-test; n=40**).

Downbound the STO frequency and IO spike frequency were inversely related. This may seem unintuitive but is explained by channel dynamics of IO cells, as more CN inhibition both increases the driving force and reduces Ca-T type inactivation, leading to more prompt spiking upon disinhibition from DCN (S5E Fig) [16,52,87]. For both the Up/Downbound zone, 5–10 Hz input lowered the STO frequency further, while at the same time increased the IO spike frequency. The lowering of the STO frequency was more pronounced in the Upbound zone. We examined the IO spike-triggered PSC to understand further how the input characteristics affected CSpk occurrences (S6C Fig). In general, for both IO unfiltered and 5–10 Hz input, IO spikes were preceded by oscillations below or at the lower range of the STO frequency (5.6 Hz for Upbound, 5.4 Hz for Downbound). For unfiltered noise, IO spikes were triggered by a small decrease in PSC amplitude, whereas for 5–10 Hz input, IO spikes were triggered by a shortening of the PSC oscillation period by 10 ms (S6C Fig). Our results suggest that, for the unfiltered input (OU input), the smaller frequencies, which are similar to the STO frequency, have a bigger impact on the system.

### Gap junction coupling drives frequency selectivity

Following our previous hypothesis that gap junctions affect IO STO synchrony, we hypothesized that this change in STO synchrony will be reflected in the loop's response to different frequencies. We observed that the frequency selectivity of the IO was lost when the gap junction was uncoupled (Fig 6A). The differences in the mean and distribution of CSpk and SSpk frequency after plasticity were much less pronounced within the zones in the uncoupled condition than in the coupled case. Furthermore, the higher IO spike frequency and lower STO frequency was completely lost for the Upbound zone, but was increased for the Downbound micromodule when the IO neurons were uncoupled (Fig 6B). The loss of the inverse STO - IO spike frequency relationship in the Upbound zone co-occurred with a loss of the oscillatory response in the IO-spike triggered PSC (Fig 6C). A higher CSpk frequency for the Downbound zone (S4B Fig) led to a less pronounced loss of the inverse STO - IO spike frequency relationship. This suggests that the gap junction coupling has an effect upstream to the PC-CN and back to the IO as there is less variation in the CN inhibitory input onto the IO.

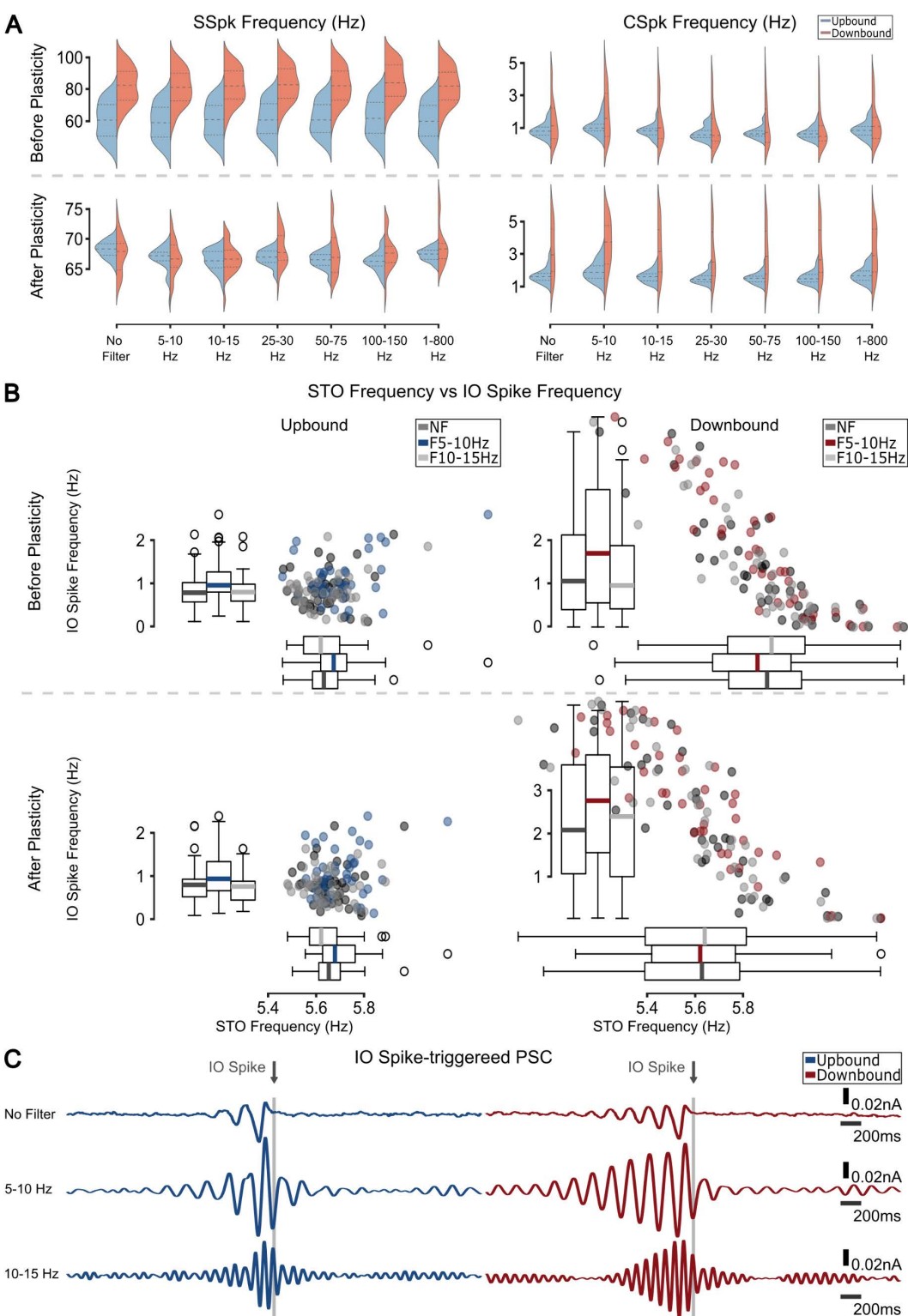

**Fig 6. Gap junction drives frequency selectivity. (A)** Firing frequency distributions for SSpks (left) and CSpks (right) before and after plasticity for different frequency bands when gap junctions are deleted. **(B)** STO frequency vs IO spike frequency for Upbound (left) and Downbound (right) zones for before (top) and after (bottom) plasticity. Downbound IO has a wider range of STO frequency and as a result a wider range of IO spikes also. **(C)** IO spike triggered PSC response. Uncoupled Upbound zones lose or have much shorter entrainment phase, while the Uncoupled Downbound still shows strong oscillatory response.

## Baseline Up/Downbound states support task-related plasticity during conditioning

Our model reproduced bidirectional plasticity in PC activity in response to arbitrary inputs (**Fig 3B**). While frozen noise may approximate inputs in the awake and behaving state, it does not reflect specific motor behaviors. To test whether the stable Up/Downbound baseline states generated by the model could support learning-related plasticity, we reproduced a classical eyeblink conditioning paradigm [88]. A conditioned stimulus (CS, 250 ms) was delivered to one PF bundle and co-terminated with an unconditioned stimulus (US, 30 ms) delivered to another PF bundle. A copy of the US was sent to the IO to elicit CF input, as in standard protocols (Fig 7A). Each block consisted of an initial US-only trial, ten CS + US pairings, and a final CS-only trial, separated by 8-12s intervals (Fig 7B).

The model captured key features of learning-related plasticity. The US elicited CSpks across the IO population (Fig 7C), and STO synchrony increased in the Upbound zone after plasticity (Fig 7D). Before learning, SSpks increased in response to the CS; this increase disappeared after learning (Fig 7E). These dynamics were consistent with the baseline states in Fig 4, as seen in before/after plasticity comparisons of SSpks, CSpks and CF pause (Fig 7F-H).

Some discrepancies with experimental data remained. We did not observe a post-learning decrease in SSpks, likely due to the absence of MLIs, which can suppress PC activity in response to increased PF input [89,90]. Nor did we observe the CS-aligned CSpk increase reported experimentally, which may depend on modulatory input from the striatum or amygdala [91], both absent from our model. However, our model did replicate the experimentally observed decrease in CSpk frequency in response to the US in the Downbound zone after learning, suggesting that this effect arises from intrinsic olivocerebellar dynamics.

## STO frequency decreases after plasticity for CS-US experiment

To better understand the learned response during CS-US trials, we examined PF weight changes over time. PF1–3 and PF5 (US) exhibited similar weight dynamics (S7A Fig), while the PF4 (CS) weight showed a pronounced decrease during plasticity, likely explaining the lack of CSpks at CS onset (Fig 7E). Trial-averaged weights (S7B Fig) confirmed this pattern: the PF4 weight declined steadily across CS trials, whereas the other PF weights remained stable. During the US, PF1–3 weights briefly decreased, then rapidly returned to baseline, while PF5 remained unaffected. Before plasticity, increased PC firing transiently accelerated STO frequency and global oscillator phase (S7C Fig), an effect abolished after plasticity.

When gap junctions were uncoupled, IO STO synchrony decreased overall but still peaked near the US (S8A Fig). SSpk responses were similar to the coupled case, but the CSpk pattern reversed: the Downbound peak increased while the Upbound decreased post-learning (S8B Fig). Uncoupling also led to higher IO firing rates (S8C Fig).

## Discussion

How brains are able to maintain acquired sensorimotor patterns while continuous plasticity is a function of reverberating activity in all elements of the system is a central problem in computational neuroscience. The olivocerebellar system is a particularly interesting case to study continuous plasticity dynamics given that many of its neurons have a high level of intrinsic activity, organized in repeated feedforward loops [4,6,7,13,19,37]. To examine the plasticity dynamics of the olivocerebellar system, we developed a biologically grounded model representing distinct, so-called Up/Downbound modules within the network [9,43,44]. We tested the model in both conditions to compare the dynamics, stability, and plasticity rules that emerge within each module. The model does not simulate the modules interacting within a shared network, but treats them as parallel, anatomically independent compartments. Our aim was to capture and contrast the intrinsic stabilization dynamics of synaptic weights under ongoing plasticity learning behaviors in the isolated olivocere-bellar system, thus keeping connectivity identical in both modules. In designing our model, we applied a reverse engi-neering approach and remained agnostic with respect to the specific functions that the cerebellar modules may compute, focusing on the closed loop dynamics of PF-PC plasticity subjected to resonances of IO spiking. We also excluded input to the IO and CN from cortical and subcortical structures and treated IO activity exclusively as a function of CN feedback

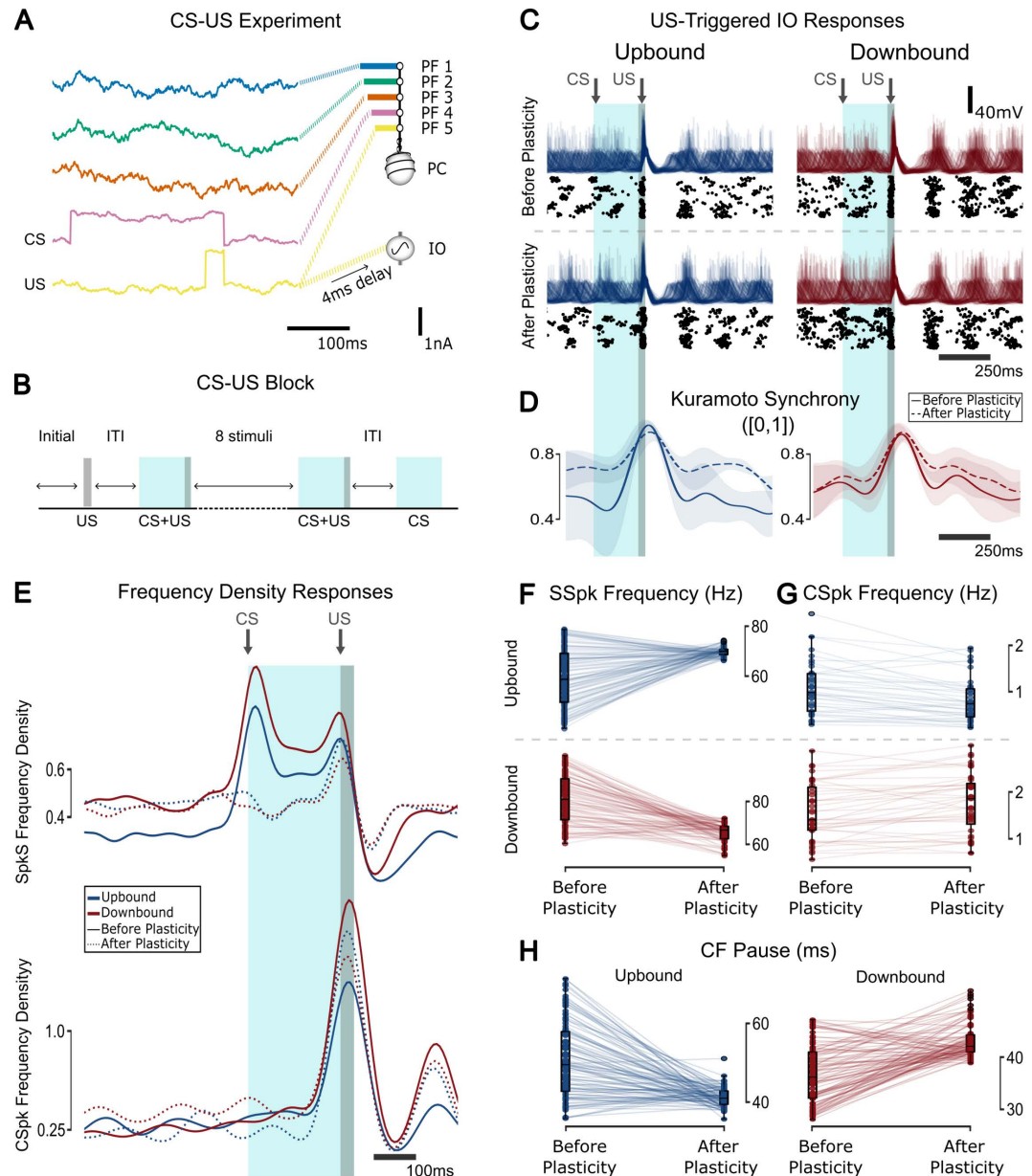

**Fig 7. Response of the olivocerebellar model to an eyeblink conditioning experiment after plasticity for Up/Downbound zones. (A)** Different PF bundles receive the CS and US inputs. **(B)** A block of the eyeblink experiment consisting of an initial US (30 ms) followed by 10 CS + US pairings (220 + 30 ms), followed by a CS (220 ms). **(C)** CSpk-triggered responses of the IO population (40 cells) membrane potential (top) and raster (bottom) for before and after eyeblink training for Up and Downbound zones. **(D)** The Kuramoto synchrony between all 40 IO cells. **(E)** CS-triggered frequency density responses of the populations for SSpk (top) and CSpk (bottom). Note that panel E shows normalized population density traces, not mean firing rates; for comparisons of firing rate distributions across zones and conditions, see panel F. **(F)** Frequency responses for SSpk (before plasticity: **t = 13.48, p < 0.001,** after plasticity: **t = 8.72, p < 0.001**, Upbound: **-8.66, p < 0.001,** Downbound: **t = 12.96, p < 0.001; two-sample Student's t-test**). The variance of SSpk frequency is significantly different after plasticity across both zones (**F = 5.87, p < 0.001; F-test using CCDF of F-distribution**) and the variance is also different for both zones across plasticity epochs (Upbound: **F = 42.01, p < 0.001**, Downbound: **F = 7.10, p < 0.001; F-test using F-distribution**). **(G)** CSpk frequency boxplot (before plasticity: **t = 3.68, p < 0.001,** after plasticity: **t = 7.39, p < 0.001; two-sample Student's t-test**). **(H)** CF Pause boxplot (before plasticity: **t = -12.78, p < 0.001,** after plasticity: **t = 5.07, p < 0.001**, Upbound: **9.68, p < 0.001**, Downbound: **t = -10.27, p < 0.001; two-sample Student's t-test**). The variance of the CF pause is significantly different after plasticity across both zones (**F = 1.83, p = 0.001; F-test using CCDF of F-distribution**) and the variance is also different for both zones across plasticity epochs (Upbound: **F = 15.62, p < 0.001**, Downbound: **F = 2.96, p < 0.001; F-test using CCDF of F-distribution**).

and intrinsic oscillatory activity. Using this network model formed by the evolutionarily conserved, essential components, we analysed resonant dynamics of the loop, PC-PF weight distributions and stability under different frequency inputs, as well as the influence of IO gap junctions on frequency responses across populations. In what follows we will discuss the main results obtained by interrogating the model; these include (1), how steady state distributions of PC activity capture directions of plasticity in Up/Downbound modules, (2) which PF input frequencies most likely evoke CSpks, (3) how synaptic weight distributions of the PFs of both modules stabilize, and (4) to what extent this stabilization depends on gap junction coupling in the IO.

## Steady state distributions of PC activity

Firing rates measured under resting conditions in awake animals likely reflect the stable physiological operating point of the olivocerebellar loop, shaped by continuously adapting plasticity mechanisms. These baseline configurations should not be interpreted as 'pre-plasticity' or naïve states, but steady states that emerge from continuous input, intrinsic excitability, and ongoing synaptic plasticity. This interpretation is supported by experimental evidence: spontaneous firing rates in Up/Downbound zones are stable over tens of seconds in adult animals [9], and zonal differences in firing rate and expression of plasticity-related markers are already present during early postnatal development [92]. These data suggest that zone-specific firing characteristics are not simply acquired through experience-dependent learning but are likely developmentally programmed and maintained through homoeostatic plasticity.

In our model, PF-PC synapses were subject to two types of plasticity, BCM and CSpk-dependent LTD. The BCM rule is a Hebbian activity dependent learning rule, and as such is expected to reflect statistical regularities in the input [93,94]. However, in a biophysically plausible setting with a large set of parameters and feedback there is no guarantee that IO and PC responses remain bounded over time. To examine weight convergence, we verified the weight values after multiple epochs of repeated, stochastic input, modeled as OU noise. We found that synaptic weight distributions and firing rates stabilized across epochs for both Up/Downbound micromodules. After an initial transient, Upbound zones displayed more weight stability across epochs, while Downbound modules showed small, bounded fluctuations around a stable mean, and thus were more sensitive to loop dynamics. In effect, ongoing plasticity tuned PC activities in the same direction as observed in physiological experiments [9,43,44] (Fig 3); thus our model showed how module-specific intrinsic PC firing rates shaped PC-PF plasticity direction. Plasticity did not simply preserve the initial intrinsic firing asymmetry, but instead reconfigured PC activity through closed-loop dynamics. Preferred activity levels emerged as a result of interactions between intrinsic excitability, ongoing input, and network feedback. In this sense, the loop does not enforce a fixed firing rate signature per module, but rather shapes zone-specific attractors under homeostatic plasticity. The intrinsic current established a rate preference that acted as a corrective "elastic force" on the weights. In the case of the Downbound module, this corrective drive was less efficient at damping weight fluctuations, leading to slower convergence and greater residual variability, whereas in the Upbound zone, convergence was faster and more stable. This leads us to predict a higher changeability of synaptic weights in Downbound zones, as the PC intrinsic rate is kept farther from circuit equilibrium in this case. This model prediction is in line with the biological observation that proteins implicated in synaptic plasticity are more prominently expressed in the Downbound zones, than those of the Upbound zones [43,44,95]. Notably, recent transcriptomics studies indicate that the subdiversity of different PC microzones is even greater than the two main categories of Up/Downbound modules, particularly in primates [42,96,97]. Hence, future models will need to consider reflecting the full heterogeneity of all PC populations.

Using our model's simulations we can conclude that while zone-specific PC characteristics are known to be intrinsic [9,42–44], the olivocerebellar loop mechanisms stabilize these differences through the interaction of BCM and CSpk-dependent long-term plasticity. However, the precise direction of baseline firing rate divergence between Up/Downbound modules may also depend on additional biological mechanisms not captured in our model, such as MLI inhibition or structured MF input.

**Input frequency dependent CSpk responses**

Changes of weight of specific PCs in our model were strongly determined by their recent input, both in terms of PF and CF input. Recent firing rate acted as a low pass proxy to history of inputs, while the CSpk timing was strongly determined by the phase state of the IO. We observed a strong preference for specific frequencies and phases of the stochastic PF input and showed that CSpks probabilities were modulated by specific temporal features of this input (Fig 4A and 4B). As the state of the IO was subject to CN population drive, we showed that the plasticity of the overall system was frequency-dependent.

Our choice of using OU PF noise [68] was motivated by emulating arbitrary input that may exhibit any frequency and any phase combination. In the long interval of our plasticity epochs (120 s), this OU input contained very different frequency compositions. This modeling strategy was chosen to reflect *in vivo* experimental data of granule cell and PF activity during learning, at rest and during walking [66,67,69,70], which show complex, high dimensional firing patterns transmitted into PCs in awake animals. The variability of the IO feedback and the resulting jitter in the latency of the CSpk response across trials, which we show in our model, has also been observed experimentally [7,85] and can be explained by phase shifts of IO STOs [52]. We analyzed the extent of the input frequency dependence by filtering inputs to include only specific frequency bands. We discovered significant differences in SSpk distributions to filtered inputs, which was particularly apparent in Downbound zones. Specifically, we found that our model responded preferentially to the 5–10 Hz frequency band, which overlapped with the IO STO frequency (5–6 Hz), indicating that specific ranges of PF frequency bands led this model closer to physiological responses.

This frequency preference suggests that IO output is maximized when PF input contains components near the intrinsic STO frequency of the IO (4–6 Hz). When STOs are aligned with this input band, IO neurons are more likely to spike, thereby increasing the probability of IO spiking. When we further investigated the relation between IO STO frequency and IO spike output, we found them to be inversely related in both Up/Downbound modules. This arose from differences in inhibitory CN feedback: stronger inhibition in the Upbound zone slowed the STOs, effectively shifting them away from the preferred frequency range and thus reducing the likelihood of IO spiking. In contrast, the Downbound zone received weaker CN inhibition, resulting in faster STOs that remained better aligned with the preferred frequency range, facilitating more frequent IO spikes. This mechanism demonstrates how CN inhibition indirectly gates plasticity through STO tuning, and aligns with *in vivo* recordings showing that CSpk rates differ across modules [7,85], with Upbound zones typically displaying lower spontaneous CSpk frequencies than Downbound zones.

**Stability of PF-PC synaptic weights**

The overarching question that motivates our model is the apparent dilemma between stability of acquired sensorimotor patterns and ever changing synaptic strengths. However, we did not specifically test the loop in a behaviorally-relevant setting, rather we examined the general case of any arbitrary input, reflecting ongoing sensorimotor behavior. Future research with our model could study input signals that more closely represent the contingencies of sensorimotor learning, with their specific frequencies and phases, related to the interaction to the body and other brain regions. Upcoming effort could focus on adding the excitatory input to the IO from the mesodiencephalic junction (MDJ) [16] and even the CN [13]; oscillatory gating of the granular layer via Golgi cells [10,98]; and/or inhibition from the MLIs [89,90]. While we deliberately excluded the excitatory CN-MDJ-IO pathway and MF collaterals to the CN to focus on the core feedback loop, we acknowledge that these inputs are likely to modulate IO excitability and contribute to zonal differences. Experimental evidence shows that excitatory input via the MDJ can evoke CSpks with much shorter latencies (4–8 ms) [15] than the slower GABAergic inhibition of over 50 ms [79,99,100], and that MF input to the CN may shift baseline excitability.

Our model was not tuned for specific outcomes but to test whether stable synaptic states could emerge from internal loop dynamics. Resonance between PF and CF activity drove frequency-dependent PF-PC synapses stabilization without

task-specific tuning. We interpret this as a baseline regime of the olivocerebellar loop, where Up/Downbound modules settle into distinct steady states under stationary input and ongoing plasticity.

We extended this framework by incorporating a classical eyeblink conditioning paradigm. The baseline states established by ongoing plasticity persisted when task-related CF inputs were introduced. During CS-US pairings, Up/Downbound modules underwent distinct learning-related adaptations, demonstrating that homeostatic stabilization provides a functional substrate for supervised, CF-driven plasticity. This reconciles experimental observations of stable baseline states with the emergence of biased plasticity during learning.

## Impact of gap junction coupling on stabilization

Gap junctions can cause synchronized IO spikes, in both Up/Downbound zones, leading to a global phase in which many individual IO neurons are coupled [16,101]. When the gap junctions were abolished in our model, the frequency selectivity of the IO response was disrupted. In the Upbound module, this disruption was accompanied by a reduction in the variability of IO spike rates and a loss of oscillatory structure in IO-triggered PSC responses. However, in the Downbound module, IO spike variability persisted to a larger degree, likely due to its intrinsically higher IO and PC firing rates and more robust CSpk activity. This differential effect underscores the zone-specific contributions of IO electrotonic coupling to timing and frequency selectivity. Moreover, this result suggested that the gap junctions have a critical influence on the frequency selectivity of the system. Indeed, when tested, we found that the frequency selectivity due to the difference in strength of the CN input, highlighted above, was lost in the absence of IO gap junction coupling. As CN inhibition turned out to not completely shape the IO STOs, this could indicate that the interplay between CN inhibition and IO gap junction coupling is at the core of this STO - IO frequency relationship. It should be noted that the interaction between GABAergic CN input to the IO and regulation of the coupling strength as well as the relevance of coupling in the IO for the prominence of its STO's has been the subject of many experimental studies. Both relations have been confirmed at the morphological, electrophysiological and pharmaceutical level, *in vivo* as well as *in vitro* [15,16,52,77–80,86,102–106]. The relationship between the level of STO's and IO frequency band is underscored by the finding that Up/Downbound modules showed different correlation strengths in this respect. Thus, as gap junction coupling facilitates IO resonance and as there are module-specific interactions, our data imply that the temporal pattern selectivity processing across Up/Downbound micromodules is enhanced by dynamic regulation of electrotonic coupling in the IO. This was further confirmed in the eyeblink conditioning paradigm: uncoupling reduced overall IO STO synchrony but preserved peak synchrony around the US. Interestingly, CSpk responses reversed post-learning Downbound increase and Upbound decrease, highlighting how coupling biases plasticity directionality. IO firing rates also increased without coupling, suggesting that electrotonic interactions constrain IO excitability and shape zone-specific plasticity. These modeling data align well with experiments in which olivary coupling is affected either genetically or pharmacologically; impairing dendrodendritic IO coupling hampers cerebellar sensorimotor learning [75].

## Limitations and biological generalization

Our model captures key internal dynamics of the olivocerebellar loop, however due to epistemic considerations several biological features were simplified or explicitly excluded from the design. The PF-PC synaptic plasticity was modeled using a BCM-inspired homeostatic rule in combination with a CSpk-triggered LTD component, while leading to LTP in the absence of CSpks. While this captures bidirectional plasticity and weight stabilization, it remains a phenomenological approximation. Experimental data on PF-PC plasticity, particularly under *in vivo* or awake conditions, suggest a large set of mechanisms that our simplified rule does not fully represent [23–27]. For instance, the CF CSpk itself was not modeled with detailed waveform dynamics, and thus the influence of CSpk duration, calcium influx or burst structure on plasticity was not explored [107,108]. In terms of input, we employed stochastic noise (OU process) to test how arbitrary

fluctuations at different frequencies shape loop dynamics and plasticity. This does not capture structured MF activity, such as bursts or sequences associated with sensory events or behavior. Nor does it reflect modulations by Golgi-Granule cell-mediated gating oscillations in the 8–12 Hz range [10,98]. However, we observed that our model responded most strongly to frequencies near 10 Hz — consistent with the filtering properties of the granular layer [98]. This suggests that Golgi cell-induced frequency gating may reinforce IO resonance at these frequencies and further constrain learning windows.

To assess whether the stable Up/Downbound configurations could support learning, we modeled a classical eyeblink conditioning paradigm typically associated with Downbound zones [5]. We applied the paradigm to both modules and found that CS-induced SSpk increases were suppressed after learning, and US-triggered CSpks decreased in the Downbound zone. Notably, these effects reversed when IO gap junctions were removed. While the conditioning paradigm captures key elements of learning-related CF modulation, it does not implement full IO-gated sensorimotor feedback loops that modulate CF activity. A recent model, incorporating MLIs, Golgi and granule cells replicated CS-induced SSpk suppression and CSpk increase during conditioning [45], but did not examine the role of IO reverberations on PC homeostatic plasticity. Our model shows that the reduced synchrony from removing gap junctions increased Downbound CSpks and reduced Upbound ones, suggesting IO coupling is required for the timing of acquired eyeblink responses [75].

The simplifications in our model were designed to probe ongoing learning properties of the resonant olivocerebellar loop, from a reverse-engineering perspective, under minimal assumptions about what is to be learned. The model captures homeostatic stabilization under persistent input, not task-driven synaptic reorganization. Future versions could test how structured task-dependent sensorimotor contingencies shape weight homeostasis, clarifying stabilization versus learning-related plasticity. Overall, our work highlights the dynamic nature of this loop, essential for understanding how the cerebellum "learns to learn".

## Methods

### Network architecture

To balance anatomical accuracy with computational tractability, we modeled each micromodule with 100 PCs, 40 CN neurons and 40 IO neurons. These ratios are grounded in known anatomical estimates: within a module, approximately 20–30 PCs converge onto a single CN neuron, while each PC projects to ~30–50 CN neurons [58]. Half of IO neurons connect to the PCs. Each of those IO neurons project to ~7–10 PCs, and each PC receives CF input from a single IO neuron [59,60]. We modeled only the inhibitory CN neurons, as these are the ones that project to the IO [53–57]. For the CN-IO synapse, we assumed a 25% reciprocal connectivity probability, resulting in each CN neuron connecting ~10 IO neurons and each IO neuron receiving from ~10 CN neurons. This structured yet simplified architecture ensures that core convergence and divergence patterns of the circuit are preserved while enabling efficient simulation (see Fig 1).

### Neuron models and justification

PCs and CN neurons were modeled using simplified adaptive exponential integrate-and-fire neurons. This abstraction was sufficient to capture the key dynamics relevant for our study: the modulation of PC firing rate by PF and CF input, and the inhibitory feedback from PCs to CN. Furthermore, IO neurons are endowed with rather unique properties. They are one of the most densely coupled via dendro-dendritic gap junctions than any other type of neuron in the mammalian brain [102,103,109] and they express high levels of voltage-gated and calcium-dependent ion-channels that allow them to oscillate at frequencies of up to 10 Hz [52,77–80]. The interplay between the excitatory and inhibitory inputs to the IO as well as the temporal relation with the subthreshold oscillations (STOs) of the coupled olivary cells determines whether and when an olivary action potential will be generated [16,64,105,110]. Because our primary focus was on IO-driven resonances and loop-level plasticity, we prioritized a more detailed multi-compartment IO model to accurately reproduce STOs

and phase-resetting phenomena. Modeling PC morphology or PC calcium dynamics was not necessary for our hypothesis (see Introduction), which depended primarily on CF timing and population-level PC output.

**Parallel fiber bundles**

We simulate the PF input, representing the activity of PF bundles, as an Ornstein-Uhlenbeck (OU) process:

$$\frac{dI^{(PF)}_{OU}}{dt} = \frac{I^{(PF)}_{OU0} - I^{(PF)}_{OU}}{\tau^{(PF)}_{OU}} + \frac{\sigma^{(PF)}_{OU}}{\sqrt{\tau^{(PF)}_{OU}}}\xi \tag{1}$$

where $\xi$ is a Gaussian white noise with mean $\mu = 0$ and standard deviation $\sigma = 1$ that scales with units of $s^{-0.5}$. The mean is $I^{(PF)}_{OU0}$ and the variance —the integral of the power spectral density— is equal to $\sigma^2$. The noise intensity is defined as $\sigma^2\tau$. Parameter values in the model are shown in Table 1.

**PC and CN neuronal models**

The PC and CN populations are modeled as adaptive exponential integrate-and-fire (AdEX) models [111]. The general cell equations for the PC and CN are described as:

$$\frac{dV}{dt} = \frac{1}{C_m}\left(-g_l(V - V_l) + g_l\Delta_T e^{\left(\frac{V-V_T}{\Delta_T}\right)} + \sum_i I^{(i)} - w\right) \tag{2}$$

$$\frac{dw}{dt} = \frac{a(V - V_l) - w}{\tau_w} \tag{3}$$

Here, Eq. 2 describes the membrane potential dynamics, while Eq. 3 shows the dynamics of the adaptation value. The $\sum_X I^{(i)}$ refers to the sum of the currents specific to cell type X (where $X \in \{PC, CN\}$ and $i$ indexes the current types affecting that cell); $\Delta_T$ is the slope factor; $V_T$ is the threshold potential; $w$ is the adaptation variable; $a$ is the adaptation coupling factor, and $\tau_w$ is the adaptation time constant. The slope factor $\Delta_T$ is a quantification of the sharpness of the spike, which can be seen as the sharpness of the sodium activation curve if the activation time constant is ignored [112]. The parameters of all AdEx cells (both CN and PC) were sampled using evenly spaced values across the listed range in Table 2. For each parameter we generated a list of linearly spaced values equal to the population size (e.g., 100 for PCs), and assigned them to individual cells without replacement. This approach ensures a uniform spread across the specified range and introduces controlled variability in intrinsic properties across the population. Hence, each cell in the population will display individual differences in firing properties. When a spike happens both the membrane potential and the adaptive variables are updated. The actual threshold potential $V_{cut} = V_T + 5\Delta_T$ is used for the neuron models. When the potential reaches $V_{cut}$ (see Table 2), a spike occurs and both the membrane potential $V$ is reset to $V_r$, and the adaptation variable $w$ is updated to the reset adaptation current $b$ and then decays until either reaching 0 or the next spike time [113].

$$@Cell\ spike: \ V = V_r;\ w+ = b \tag{4}$$

**Table 1. Parameters of the Ornstein-Uhlenbeck processes.**

| Parameters | Baseline current $I_{OU0}$ (nA) | OU time constant $\tau_{OU}$ (ms) | OU standard deviation $\sigma_{OU}$ (nA) |
|---|---|---|---|
| PF | 1.3 | 50 | 0.25 |
| IO | -0.3 | 50 | 0.3 |

**Table 2. Parameters of the AdEx cell types.**

| Parameters | PC | CN |
|---|---|---|
| Membrane capacitance $C_m$ (pF) | $75 \pm 1.0[N_{PC}]$ | $281 \pm 1.0[N_{CN}]$ |
| Leakage conductance $g_l$ (nS) | $30 \pm 1.0[N_{PC}]$ | $30 \pm 1.0[N_{CN}]$ |
| Leakage reversal potential $V_l$ (mV) | -70.6 $\pm 0.5[N_{PC}]$ | -70.6 $\pm 0.5[N_{CN}]$ |
| Threshold potential $V_T$ (mV) | -50.4 $\pm 0.5[N_{PC}]$ | -50.4 $\pm 0.5[N_{CN}]$ |
| Slope factor $\Delta_T$ (mV) | $2 \pm 0.5[N_{PC}]$ | $2 \pm 0.5[N_{CN}]$ |
| Adaptation time constant $\tau_w$ (ms) | $144 \pm 2.0[N_{PC}]$ | $30 \pm 1.0[N_{CN}]$ |
| Adaptation coupling factor $a$ (nS) | $4 \pm 0.5[N_{PC}]$ | $4 \pm 0.5[N_{CN}]$ |
| Reset potential $V_r$ (mV) | -70.6 $\pm 0.5[N_{PC}]$ | -65 $\pm 0.5[N_{CN}]$ |
| Reset adaptation current $b$ (nA) | $0.0805 \pm 0.001[N_{PC}]$ | $0.0805 \pm 0.001[N_{CN}]$ |
| Intrinsic current $I_{int}$ (nA) | $0.35 \pm 0.21[N_{PC}]$ | 1.2 |

Our goal for using the randomization of parameters is to have variability across the cells in each population. Since the CN is inhibited by the PC we will already have enough variability in the inhibitory current as each PC is different. Also, the CN cells have their own variability across the parameters. Hence, we do not randomize the intrinsic current $I_{int}$ of the CN as we can then see the direct effect of one PC onto the CN.

### Purkinje cell Currents

For the PC, there are 2 currents denoted $I^{(PC)}_{int}$ and $I_{PF}$. The former is the activation (intrinsic) input current given to the cell to tune the desired intrinsic firing frequency distribution. The latter is the postsynaptic current (PSC) input entering the PC soma from the PFs (Fig 1A).

$$\sum I_{PC} = I_{PF} + I^{(PC)}_{int}$$

(5)

### Cerebellar nuclei Synaptic Currents

The CN currents are shown in Eq. 6, where $I^{(CN)}_{int}$ is the intrinsic current of the CN and $I^{(CN)}_{PC}$ is the inhibitory current from the PC to the CN. It is a filtered version of the PC spike train that decays exponentially in the absence of spikes. Upon each PC spike, $I^{(CN)}_{PC}$ is updated by a constant $\gamma^{(CN)}_{PC}$ (see Table 6). This means that the total current $I_{int} - I^{(CN)}_{PC}$ will be smaller and the CN will spike less (inhibition from PC).

$$\sum I_{CN} = I^{(CN)}_{int} - I^{(CN)}_{PC}$$

(6)

### Inferior olive model

The IO model is based on the De Gruijl model [108] —a modification of the Schweighofer model [87] with an added axonic current. The general cell equations for the model are shown below:

$$C_m \frac{dV_s}{dt} = -\sum I^{(s)} + I^{(s)}_{app} + I^{(IO)}_{OU} + I^{(IO)}_{CN}$$

(7)

$$C_m \frac{dV_d}{dt} = -\sum I^{(d)} + I^{(d)}_{app}$$

(8)

$$C_m \frac{dV_a}{dt} = -\sum I^{(a)} \tag{9}$$

$$\frac{dI^{(IO)}{}_{OU}}{dt} = \frac{I^{(IO)}{}_{OU0} - I^{(IO)}{}_{OU}}{\tau^{(IO)}{}_{OU}} + \frac{\sigma^{(IO)}{}_{OU}\xi}{\sqrt{\tau^{(IO)}{}_{OU}}} \tag{10}$$

Where, the membrane potential specific to each compartment, somatic s, dendritic d and axonic a is represented with $V_i$ (where $i = s, d, a$). The equations for the ionic conductances of the general cell are shown in Table 3 and their parameters in Table 4. All parameters are randomized in the same way as for the AdEx parameters. The IO receives an a noise input $I_{IO_{OU}}$ modeled as an OU process (similarly to the PF input, see Table 1). With this input we model the baseline firing rate of the IO as well as modeling inputs from other brain regions. Furthermore, the inhibitory input from the CN onto the IO is modeled by the current $I^{(IO)}{}_{CN}$, as described in the Synapses subsection. Finally, two applied currents $I^{(i)}{}_{app}$ (where $i = s, d$) are in the dendrites and somatic compartments. These are used when adding external current into the cell. The dynamics of each activation or inactivation variable (see Table 5) is given by the differential equation below. In this equation $z$ indicates the variables $h, k, l, n, x, q, r, s, h_a$ and $x_a$. There are 2 exceptions, $m$ and $m_a$ which have a small enough $\tau$ that they are approximately equal to their steady-states ($m_\infty$ and $m_{\infty a}$, respectively).

$$\frac{dz}{dt} = \frac{(z_\infty - z)}{\tau_z} \tag{11}$$

Three morphological parameters determine the electrotonic properties of the three-compartmental model: the ratio of the somatic area to the total somatic and dendritic surface area $p$; the ratio of the somatic area to the total somatic and axonic surface area $p_2$, and the electrotonic coupling conductance between the 3 compartments. The current flowing from the somatic into the dendritic compartment $I_{sd}$ and from the dendritic into the somatic compartments $I_{ds}$, as well as the leakage currents for the soma $I_{ls}$ and for the dendrites $I_{ld}$ are represented in this model [87]. We also have the current flowing out from the soma into the axon $I_{sa}$ and from the axon into the soma $I_{as}$ as well as the leakage for the axonic compartment $I_{la}$ [108]. Furthermore, the dynamics of the calcium concentration $\left[Ca^{2+}\right]$ is defined as follows:

$$\frac{d\left[Ca^{2+}\right]}{dt} = -3.0 I_{Ca_h} - 0.075 \left[Ca^{2+}\right] \tag{12}$$

**Table 3. Ionic conductances of the IO cell.**

| Somatic currents $\sum I_s$ | Dendritic currents $\sum I_d$ | Axonic currents $\sum I_a$ |
|---|---|---|
| $I^{(s)}{}_{Na} = g_{Na} m_\infty{}^3 h (V_s - V_{Na})$ | $I^{(d)}{}_{Ca_h} = g_{Ca_h} r^2 (V_d - V_{Ca})$ | $I^{(a)}{}_{Na} = g_{Na_a} m_\infty{}^3 h_a (V_a - V_{Na})$ |
| $I^{(s)}{}_{Ca_l} = g_{Ca_l} k^3 l (V_s - V_{Ca})$ | $I^{(d)}{}_{K_{Ca}} = g_{K_{Ca}} s (V_d - V_K)$ | $I^{(a)}{}_{K_a} = g_{K_a} x_a{}^4 (V_a - V_K)$ |
| $I^{(s)}{}_{K_{dr}} = g_{K_{dr}} n^4 (V_s - V_K)$ | $I^{(d)}{}_c = g_c (V_d - V_{de}) f (V_d - V_{de})$ | $I^{(a)}{}_{la} = g_{la} (V_a - V_l)$ |
| $I^{(s)}{}_{K_s} = g_{K_s} x_s{}^4 (V_s - V_K)$ | $I^{(d)}{}_{ld} = g_{ld} (V_d - V_l)$ | $I^{(a)}{}_{sa} = \frac{g_{int}}{p_2} (V_a - V_s)$ |
| $I^{(s)}{}_h = g_h q (V_s - V_h)$ | $I^{(d)}{}_{sd} = \frac{g_{int}}{(1-p)} (V_d - V_s)$ | |
| $I^{(s)}{}_{ls} = g_{ls} (V_s - V_l)$ | | |
| $I^{(s)}{}_{ds} = \frac{g_{int}}{p} (V_s - V_d)$ | | |
| $I^{(s)}{}_{as} = \frac{g_{int}}{(1-p_2)} (V_s - V_a)$ | | |

**Table 4. Parameters of the IO Standard Cell.**

| Conductances (mS/cm²) | | Reversal Potentials (mV) | |
|---|---|---|---|
| $g_{Na}$ | $150 \pm 1.0[N_{IO}]$ | $V_{Na}$ | $55 \pm 1.0[N_{IO}]$ |
| $g_{Ca_l}$ | $1.4 \pm 0.05[N_{IO}]$ | $V_{Ca}$ | $120 \pm 1.0[N_{IO}]$ |
| $g_{K_{dr}}$ | $9.0 \pm 0.1[N_{IO}]$ | $V_K$ | $-75 \pm 1.0[N_{IO}]$ |
| $g_{K_s}$ | $5.0 \pm 0.1[N_{IO}]$ | $V_h$ | $-43 \pm 1.0[N_{IO}]$ |
| $g_h$ | $0.12 \pm 0.01[N_{IO}]$ | $V_l$ | $10 \pm 1.0[N_{IO}]$ |
| $g_{ls}$ | $0.017 \pm 0.001[N_{IO}]$ | Membrane Capacitance ($\mu$F/cm²) | |
| $g_{Ca_h}$ | $4.5 \pm 0.1[N_{IO}]$ | $C_m$ | $1.0 \pm 0.1[N_{IO}]^2$ |
| $g_{K_{Ca}}$ | $35 \pm 0.5[N_{IO}]$ | Cell Morphology | |
| $g_{ld}$ | $0.016 \pm 0.001[N_{IO}]$ | $g_{int}$ | $0.13 \pm 0.001[N_{IO}]$ |
| $g_{Na_a}$ | $240 \pm 1.0[N_{IO}]$ | $p$ | $0.25 \pm 0.01[N_{IO}]$ |
| $g_{K_a}$ | $240 \pm 0.5[N_{IO}]$ | $p_2$ | $0.15 \pm 0.01[N_{IO}]$ |
| $g_{la}$ | $0.017 \pm 0.001[N_{IO}]$ | | |

**Table 5. Steady-state activation/inactivation and time constants of IO ionic conductances.**

| $I$ | Steady-state Activation/Inactivation | Time constant $\tau$ (ms) | Forward rate function $\alpha$ | Backward rate function $\beta$ |
|---|---|---|---|---|
| $I_{Na}$ | $m_\infty = \frac{1}{1+e^{-\left(\frac{30+V_s}{5.5}\right)}}$ | | | |
| | $h_\infty = \frac{1}{1+e^{\frac{70+V_s}{5.8}}}$ | $\tau_h = 3e^{-\left(\frac{40+V_s}{33}\right)}$ | | |
| $I_{Ca_l}$ | $k_\infty = \frac{1}{1+e^{-\left(\frac{61+V_s}{4.2}\right)}}$ | $\tau_k = 5$ | | |
| | $l_\infty = \frac{1}{1+e^{\frac{85.5+V_s}{8.5}}}$ | $\tau_l = 35 + \frac{20e^{\left(\frac{160+V_s}{30}\right)}}{1+e^{\left(\frac{84+V_s}{7.3}\right)}}$ | | |
| $I_{K_{dr}}$ | $n_\infty = \frac{1}{1+e^{-\left(\frac{3+V_s}{10}\right)}}$ | $\tau_n = 5 + 47e^{\left(\frac{50+V_s}{900}\right)}$ | | |
| $I_{K_s}$ | $x_{\infty s} = \frac{\alpha_{x_s}}{\alpha_{x_s}+\beta_{x_s}}$ | $\tau_{x_s} = \frac{1}{\alpha_{x_s}+\beta_{x_s}}$ | $\alpha_{x_s} = \frac{0.13(V_s+25)}{1-e^{-\left(\frac{25+V_s}{10}\right)}}$ | $\beta_{x_s} = 1.69e^{-0.0125(35+V_s)}$ |
| $I_h$ | $q_\infty = \frac{1}{1+e^{\frac{80+V_s}{4}}}$ | $\tau_q = \frac{1}{e^{-(14.6+0.086V_s)}+e^{-(1.87+0.07V_s)}}$ | | |
| $I_{Ca_h}$ | $r_\infty = \frac{\alpha_r}{\alpha_r+\beta_r}$ | $\tau_r = \frac{5}{\alpha_r+\beta_r}$ | $\alpha_r = \frac{1.7}{1+e^{-\left(\frac{5+V_d}{13.9}\right)}}$ | $\beta_r = \frac{0.02(V_d+8.5)}{e^{\left(\frac{8.5+V_d}{5}\right)}-1}$ |
| $I_{K_{Ca}}$ | $s_\infty = \frac{\alpha_s}{\alpha_s+\beta_s}$ | $\tau_s = \frac{1}{\alpha_s+\beta_s}$ | $\alpha_s = min\left(\left[Ca^{2+}\right]_0, 0.01\right)$ $\left[Ca^{2+}\right]_0 = 2 \cdot 10^{-5}\left[Ca^{2+}\right]$ | $\beta_s = 0.015$ |
| $I_{Na_a}$ | $m_{\infty a} = \frac{1}{1+e^{-\left(\frac{30+V_a}{5.5}\right)}}$ | | | |
| | $h_{\infty a} = \frac{1}{1+e^{-\left(\frac{60+V_a}{5.8}\right)}}$ | $\tau_{h_a} = 1.5e^{-\left(\frac{40+V_a}{33}\right)}$ | | |
| $I_{K_a}$ | $x_{\infty a} = \frac{\alpha_{x_a}}{\alpha_{x_a}+\beta_{x_a}}$ | $\tau_{x_a} = \frac{1}{\alpha_{x_a}+\beta_{x_a}}$ | $\alpha_{x_a} = \frac{0.13(V_a+25)}{1-e^{-\left(\frac{25+V_a}{10}\right)}}$ | $\beta_{x_a} = 1.69e^{-0.0125(35+V_a)}$ |

Two distinct types of calcium currents exist in the IO neuron: a low-threshold current $I_{Ca_l}$ located in the soma and a high-threshold current $I_{Ca_h}$ located in the dendrites [77,114]. As the window of conductance of $I_{Ca_l}$ is around the resting membrane potential, the IO cell is excited in response to hyperpolarizing currents [77,114]. On the other hand, $I_{Ca_h}$ is non-inactivating. Thus, a depolarizing dendritic input results in a prolonged plateau potential. As calcium enters the

**Table 6. Baseline synaptic parameters and delays.**

| Synapses | Parameters | Values |
|---|---|---|
| PF-PC | Moving threshold $\theta_{M0}$ | 60 Hz |
| | Threshold time constant $\tau_M$ | 15 ms |
| | CSpk induced LTD constant $A_{CSpk}$ | -0.01 nA |
| | CSpk induced LTD time constant $\tau_{CSpk}$ | 350 ms |
| PC-CN | Inhibitory current from PC to CN $\gamma^{(CN)}_{PC}$ | 0.004 nA |
| | Synaptic delay | 10 ms |
| CN-IO | Inhibitory current from CN to IO $\gamma^{(IO)}_{CN}$ | -1.8 nA |
| | $IO_{CN}$ time constant $\tau^{(IO)}_{CN}$ | 30 ms |
| | Synaptic delay | 50 ms |
| IO-IO | Coupling conductance $g_c$ | 0.00125 mS/cm$^2$ |
| IO-PC | Inhibitory current from IO to PC $\gamma^{(PC)}_{IO}$ | 0.22 nA |
| | $I^{(PC)}_{IO}$ time constant $\tau^{(PC)}_{IO}$ | 30 ms |
| | Synaptic delay | 15 ms |

dendrites, the calcium-activated potassium current $I_{K_{Ca}}$ is activated. This current abruptly terminates the plateau potential after about 30 milliseconds [77,114]. Due to the long time constant (several hundred milliseconds) of $I_{K_{Ca}}$, there is a long afterhyperpolarization which, in turn, de-inactivates $I_{Ca_l}$ resulting in a postinhibitory rebound spike. This current contributes to the subthreshold oscillations as it contributes to amplitude and frequency [87,115]. Furthermore, somatic sodium spikes are generated with the sodium current $I_{Na}$. These are terminated by an outward delayed rectifier potassium current $I_{K_{dr}}$ [77,114]. Finally, the axon hillock includes a sodium current $I_{Na_a}$ (whose inactivation function was altered to allow for the generation of bursts of spikes) and potassium current $I_{K_a}$ [108].

### Synapses

Table 6 shows the parameters and synaptic delays for each synapse in the model. The synaptic delay and time constants were chosen from literature for the PC to CN and the CN to IO delays [100,108,116–119]. Other parameters were selected to reproduce the electrophysiological results shown in Fig 3. Synaptic parameters in Table 6 were held constant across simulations to isolate the effects of plasticity mechanisms and network architecture. While neuronal parameters in Tables 2 and 4 were varied to reflect known biological heterogeneity, fixed synaptic parameters ensured consistent scaling of synaptic inputs, allowing clearer interpretation of plasticity-driven dynamics. When applicable, these values are overridden by those in Table 7 for zone-specific simulations. While the synaptic values chosen fall within biologically plausible ranges, we manually verified that the key model outcomes —including the frequency preference for CSpk generation and PF-PC weight stabilization— were robust to variation in these delay parameters.

**Table 7. Parameter values for zone-specific simulations of micromodules.**

| Zone | Coupling | Synaptic currents | | Conductance | IO OU current parameters | |
|---|---|---|---|---|---|---|
| | | CN to IO $\gamma^{(IO)}_{CN}$ (nA) | PC to CN $\gamma^{(CN)}_{PC}$ (nA) | $g_{Cal}$ (mS/cm$^2$) | $I^{(IO)}_{OU_0}$ (nA) | $\sigma^{(IO)}_{OU}$ (nA) |
| Upbound | Coupled | -2.0 | 0.005 | 1.2 | -0.7 | 0.7 |
| | Uncoupled | -2.0 | 0.005 | 1.4 | -0.3 | 0.7 |
| Downbound | Coupled | -1.8 | 0.004 | 1.4 | -0.3 | 0.3 |
| | Uncoupled | -1.8 | 0.004 | 1.4 | -0.6 | 0.7 |

## PF-PC connectivity

Each PC receives input from 5 PF bundles with different (meta)synaptic weights. These weights are sampled from the uniform distribution and normalized to a sum of 5. This resembles a draw from a scaled symmetric Dirichlet distribution with concentration parameter $\alpha > 1$, i.e., the values are favored to the center of the simplex. The PSC $I_{PF}$ (see PC model) is calculated as:

$$I_{PF} = \frac{1}{5}\sum_j^{N_{PF}} I_j w_j \tag{13}$$

## PF-PC Plasticity

The plasticity mechanism between the PF bundles and PCs consists of an intrinsic homeostatic mechanism $w_{BCM}$ and a long-term depression (LTD) mechanism $w_{CSpk}$. The homeostatic mechanism is based on the Bienenstock-Cooper-Munro-type (BCM) plasticity mechanism [120]. LTD happens at the IO spike, whereas LTP happens during the absence of IO spikes. The final weight is as shown in Eq. 14. Where, $i$ is the PC (where $i = 1, ..., 100$) and $j$ is the synapse of that PC (where $j = 1, ..., 5$).

$$w_{i,j} = w_{BCM_{i,j}} + w_{CS_{i,j}} \tag{14}$$

## BCM

The homeostatic component of synaptic plasticity is governed by a modified BCM rule. This formulation builds on the original BCM model [120] but introduces additional damping to match cerebellar PC firing ranges.

Let $\rho_{PF_j}$ denote the effective firing rate (Hz) of the $j$-th PF bundle, computed from the amplitude of the OU-driven synaptic current, and let $\rho_{PC_i}$ be the moving average firing rate (Hz) of the $i$-th PC over a 200 ms window. The sliding plasticity threshold $\theta_{M_i}$ (Hz) for the $i$-th PC evolves as shown in Eq. 16.

$$\frac{dw_{BCM_{i,j}}}{dt} = \rho_{PF_j}\varphi\left(\rho_{PC_i}, \theta_{M_i}\right) \tag{15}$$

$$\frac{d\theta_{M_i}}{dt} = \left[\left(\rho_{PC_i}\right)^2 - \frac{\theta_{M_i}}{\tau_M}\right] \tag{16}$$

where, $\tau_M$ is the threshold time constant. The effective PF firing rate, $\rho_{PF}$, is computed by transforming the instantaneous synaptic current ($I_{PF}$, in nA) into units of Hz by scaling with the amplitude of the OU process and assuming a linear mapping between current magnitude and presynaptic rate. This transformation is valid under the model assumption that PF input represents the activity of a large bundle of asynchronously active fibers whose summed conductance mimics a continuous input current. Moreover, $\varphi_i$ represents the general change in plasticity specific to the $i$-th PC, as it depends on the firing rate $\rho_{PC_i}$, and the sliding threshold $\theta_{M_i}$, both of which vary across PCs. This term is then modulated by $\rho_{PF_j}$ for each PF bundle:

$$\phi_i = \frac{\rho_{PC_i}\left(\rho_{PC_i} - \theta_{M_i}\right)}{\theta_{M_i}} \tag{17}$$

This classical parabolic term (Eq. 17) has zeros at $\rho_{PC} = 0, \theta_M$ and a minimum of $\varphi = -\frac{1}{4}\theta_M$ at $\rho_{PC} = \frac{1}{2}\theta_M$. However, left unbounded, this can lead to unrealistically large potentiation at high firing rates. To limit this, we introduced a damping factors:

$$\frac{dw_{BCM_{i,j}}}{dt} = 5\rho_{PF_j}\tanh\left(0.01\frac{\rho_{PC_i}\left(\rho_{PC_i} - \theta_{M_i}\right)}{\theta_{M_i}}\right) \tag{18}$$

where $c = 0.01$ is a shape constant. In implementation (Brian2), we ensure the *tanh* argument is dimensionless by multiplying the rate term by a time unit (second). This hyperbolic term flattens the curve at high firing rates, ensuring that potentiation and depression remain within physiologically plausible bounds.

This formulation yields bidirectional weight changes depending on the position of $\rho_{PC}$ relative to $\theta_M$, with potentiation when $\rho_{PC} < \theta_M$ and depression when $\rho_{PC} > \theta_M$. The *tanh* function ensures saturation and boundedness of weight change. The only hard constraint is imposed on $\rho_{PC}$, limited to 250 Hz to prevent pathological firing, but no explicit bounds are placed on $w_{BCM_j}$.

To assess the robustness of the BCM mechanism, we varied the threshold time constant, a key parameter controlling the speed at which the PC activity influences synaptic change. The overall behavior of the model remained robust as the PF-PC weights stabilized but the Up/Downbound modules diverged in their plasticity outcomes only for a narrow range of time constant values. Namely, net depression in Downbound modules and net potentiation in Upbound modules. Outside this range, homeostasis was still achieved, but the direction of synaptic change was not aligned with the expected biological behavior. These results suggest that, while the system operates in a resilient homeostatic regime, certain features of learning dynamics are parameter-sensitive and emerge only within a constrained space of biologically plausible plasticity kinetics.

## LTD vs LTP

LTD is modeled as a negative change of the synaptic weight proportional to the activity of the PF and with a time decay of 350 ms (Table 7) simulating the calcium release time (Fig 2D). Upon each IO spike, the LTD term was incremented by a constant by a constant $A_{CSpk}$ is scaled by the absolute deviation from the instantaneous PF input $\rho_{PF_j}$ (Hz) from its mean level $I_{PF_0}$ (converted to Hz by explicitly multiplying the appropriate unit scaling factors). As a consequence, LTP occurs in periods without IO spikes.

$$\frac{dw_{CS_{i,j}}}{dt} = \frac{-w_{CS_{i,j}}}{\tau_{CSpk}} \tag{19}$$

$$@IO\ spike: \ w_{CS_{i,j}} += \ A_{CSpk} \left| \rho_{PF_j} - I_{PF_0} \right| \tag{20}$$

As the CSpk-triggered LTD component was incremented by a negative amount and decays exponentially back toward zero between IO spikes, it always remains negative. This decay can allow the BCM-driven component $w_{BCM_{i,j}}$ to dominate when its positive contribution exceeds $w_{CS_{i,j}}$. The total synaptic weight change is the sum of these two contributions.

## PC-CN Synapse

The inhibitory synapse from PC to the CN is modeled by a discrete increase in $I^{(CN)}_{PC}$ at the time of a presynaptic PC spike. After each increase, $I^{(CN)}_{PC}$ decays exponentially back toward zero, resulting in a temporary suppression of CN firing.

$$\frac{dI^{(CN)}_{PC}}{dt} = \frac{-I^{(CN)}_{PC}}{\tau^{(CN)}_{PC}} \tag{21}$$

$$@PC\ spike: \ I^{(CN)}_{PC} += \ \gamma^{(CN)}_{PC} \tag{22}$$

Each PC projects to 16 CN cells and each CN cell receives on average from 40 PCs, ranging from 30 to 52 (Fig 1A). Each PC source indices is repeated for each PC and CN target indices are sampled without replacement from the CN population.

## CN-IO Synapse

Analogously to the PC-CN synapse, the new synaptic current $I^{(IO)}{}_{CN}$ increases with CN spikes. The decay time of $I^{(IO)}{}_{CN}$ is small (30 ms), and enters the IO at the soma ($I^{(S)}{}_{app}$). This inhibitory input can be seen as the asynchronous release of GABA onto the IO cell [99]. The update value of $\gamma^{(IO)}{}_{CN}$ is seen in Table 6 and the synaptic update is done as follows:

$$\frac{dI^{(IO)}{}_{CN}}{dt} = \frac{-I^{(IO)}{}_{CN}}{\tau^{(IO)}{}_{CN}} \tag{23}$$

$$@CN\ spike : \ I^{(IO)}{}_{CN}+ = \ \frac{\gamma^{(IO)}{}_{CN}}{N_{CN}} \tag{24}$$

Each CN projects to 10 IO cells and each IO cell receives on average from 10 CN cells, ranging from 6 to 16 (Fig 1A). The connectivity is generated ensuring these ratios. CN cell indices are repeated for each CN cell, while IO indices are sampled without replacement from the IO population, avoiding duplicate connections.

## IO-IO Gap Junction

The electrotonic coupling between the cells is represented by a current $I_c$ (see Table 6) in the dendrites and the transjunctional voltage dependence of the gap junction conductance $f$ [87] is defined below. In the coupled case, each IO neuron connects to all other neurons in the population (Fig 1A). The transjunctional voltage dependence of the gap junction conductance (Eq. 26) is defined as $V = V_d - V_{de}$, where both $V_d$ and $V_{de}$ are dendritic membrane potentials.

$$I_c = g_c\,(V_d - V_{de})\,f\,(V_d - V_{de}) \tag{25}$$

$$f(V) \ = \ 0.6e^{\frac{v^2}{50^2}} + 0.4 \tag{26}$$

## IO-PC Synapse

The modeling of the IO-PC synapse focuses on the pause mechanism following a CSpk. Each PC receives a CF input from one IO as seen in Fig 1A. As the PC AdEX model already has an adaptation variable $w$ that increases the interspike interval (ISI), the synaptic modeling only requires the increase of $w$ following a presynaptic IO spike. The update of the $w$ adaptation variable is increased as shown in Eq. 27. As $w$ is increased to a higher value, the PC cannot spike until $w$ decays to a lower value.

$$@IO\ spike : \ w^{PC}+ = \ \gamma^{(PC)}{}_{IO} \tag{27}$$

Only half of the IOs synapse with PCs. Each of these IO CF connects to on average 5 PCs (ranging 2–9) and each PC receives from only 1 IO cell (Fig 1A). IO cell indices are sampled without replacement from the IO population. Each PC is either assigned a unique IO source, or —when convergence exceeds the IO population size— assigned an IO neuron randomly selected from a pre-sampled pool, ensuring the intended level of convergence while avoiding duplicate connections.

## Upbound and Downbound zones

To reproduce the physiological results of the different zones we did a grid search over 2 synaptic currents, 1 IO conductance and the IO OU current (see Table 7). This grid search was done in steps of 0.1 for all parameters except for the PC to CN synaptic current $\gamma^{(CN)}{}_{PC}$ which was done in steps of 0.001. Then we tried to find a set of parameters such that we

could reproduce physiological firing ranges with minimal parameter changes between cases. Table 7 shows the different parameters that need to be changed to reproduce electrophysiological results as shown in Fig 3.

The values of Table 7 override the corresponding baseline parameters listed in Table 6 for the indicated zone and coupling condition. These variations reflect biologically plausible differences in inhibitory strength, dendritic conductance, and IO noise that give rise to module-specific dynamics. Parameters not listed here retain their default values from Table 6.

### Filtered inputs

Filtered inputs were obtained via bandpass filtering. For each noise input and frequency range ($f_{clow}$ to $f_{chigh}$), a forward 6th butterworth filter was applied on the mean-offset removed original noise signal, obtaining $y_{filtered}$. Filtering was done as cascaded second-order sections, taking into account the rate at which the noise was sampled as $\left(f_s = \frac{1}{dt}\right)$ with $dt$ being the time step of the simulation. To account for power loss due to filtering, a correction was applied by multiplying the frequency-domain energy:

$$I_{PF_{filtered}} = y_{filtered} \frac{\sqrt{\sum |Y_{unfiltered}(f)|^2}}{\sqrt{\sum |Y_{filtered}(f)|^2}} + I_{PF_0}$$

(28)

### CS-US Experiment

We used step currents of different amplitude and duration (Table 8) to simulate the CS and US. The US was copied onto the IO with a 4ms delay and multiplied by 2.8 to have the desired effect on the IO spike response. The ITI was randomly chosen between 8–12 seconds.

### Simulations

For all simulations the Euler-Maruyama method was used to simulate the frozen noise. For the rest of the neuron groups, the forward Euler method was used to integrate the differential equations. Two integration timesteps were used, one for the simulations (0.025 ms) and one for the recording of the neuron output (1 ms). All experiments had a simulation length of 120 s. We first started by generating a seed which included generating a network with connectivities, cell parameters and frozen PF input. When modeling the different micromodules we used the parameters shown in Table 7.

The "before plasticity" condition was used to initialize the network and is meant to represent a biologically plausible but non-stationary starting configuration. It does not reflect a steady-state physiological condition. Physiological comparisons should instead be made using the "after plasticity" condition, which represents the stable operating point of the model under continuous adaptation.

The frozen PF input was used for the "no plasticity" experiment. For the subsequent "plasticity" epoch, we enabled the BCM and LTD mechanisms and applied the same frozen PF input. At the end of each plasticity epoch, the evolving PF-PC synaptic weights were smoothed using a Savitzky–Golay filter (window size = 100 samples, polynomial order = 2) to reduce variability caused by the instantaneous state of the LTD re LTP mechanism. Without smoothing, the final weight could be biased if the simulation ended immediately after a CSpk, when the LTD term is transiently active. The final smoothed

**Table 8. Parameter values for the different micromodules.**

| Stimulus | Pulse amplitude (nA) | Pulse duration (ms) |
|---|---|---|
| CS | 1.5 | 250 |
| US | 1.5 | 30 |

weight value was then used as the static weight for the subsequent "after plasticity" simulation with the same frozen input. This procedure was repeated for the second, third and fourth plasticity epochs (see Fig 2G-L).

The experiments with filtered signals were done in the same way. The only difference was that the new PF input was a filtered version of the same frozen input. We still had the same network seed. We redid the experiments for the following frequency bands: 5–10 Hz, 10–15 Hz, 15–20 Hz, 25–30 Hz, 50–75 Hz, 100–150 Hz and 1–800 Hz (see Fig 5A).

The CS - US experiment (light/puff) was done by adding a step current of 250 ms to the PF 4 input which represented the CS. The same was done for PF 5 but with a 30 ms step current input that represented the US. This input was also copied (with a 4 ms delay) to the IO to elicit CSpks. The two signals co-terminated. A block consisted of an initial US followed by 10 CS-US stimuli with varying ITIs and it finished with a final CS.

For each of the simulations we also did a copy simulation with the uncoupled scenario. This is the same seed, meaning that we had the same network, same connectivity, same cell parameters, and same frozen PF input. The only difference was that the coupling conductance $g_c$ (see Table 3) of the coupling current $I_c$ is lowered to 0 ($g_c = 0$).

## Quantification of firing rates

All reported firing rates (SSpk, CSpk, IO, CN) were computed using 1-second time bins for the entire experiment time. Spike trains were binned, converted to firing rates (Hz), and averaged across all recorded cells in the given population. For statistical comparisons between plasticity conditions, outlier exclusions and variance thresholding were performed using interquartile range filtering (5–95th percentiles), as described in the code and consistent across figures.

## Code

All code used for simulations and analysis is available at: https://github.com/eliasmateo95/CerebellarLoop.

## Statistical Analysis

To assess differences across simulation conditions, we used two types of parametric statistical tests depending on the data distribution. Two-sample Student's *t*-tests were applied for comparisons of means across groups. F-tests were used to compare variances across groups. All tests were two-sided and group sized ($n = 40$ for CN and IO cells, $n = 100$ for PCs) are reported in figure captions.

## Supporting information

**S1 Fig. Gap junctions increase spiking and subthreshold synchrony (A) Gap junctions are disabled by setting the gap junction current = 0.** For the Downbound micromodule, we compare the IO membrane response (B), raster (C), and correlation matrix of membrane potentials (D) in coupled (top) and uncoupled networks (bottom). Synchrony directly impacts the correlations in weight changes expected at the PF-PC synapse.
(TIF)

**S2 Fig. Synchrony and Clustering.** In a single micromodule we observe a few clusters in the short time scale whose stability fades exponentially. (A) Kuramoto synchrony metric for STOs is derived from IO Vm in 5 transformation steps. (B) Stability of global oscillator (i.e., usability as a low dimensional global clock): global phase (mean Hilbert phase of all IO cells) from start to end of an 80 second period. The deviation from horizontal indicates phase drifts across phase in radians. Both Upbound and Downbound IO have small drifts and maintain a robust global phase, equating with a single synchronized cluster. (C) Delta phase plot. Difference of phase of each IO neuron to the global phase over time. Plotted modulo 32pi radians to preserve space. Phase-locked oscillators are clearly visible as overlapping lines, while lag/leap events are shown as vertical jumps. Comparing long (D, 30s) and short term (E, 1s) clusterization with the pairwise

kuramoto phase reveals a few clusters but a single global oscillator in the 30s frame. (F) Voltage trace of the clusters found in E. The black bars indicate the 1s used for clustering. (G) Cluster stability of 1s clusters. In intervals of 10ms, 1s clusters were found with HDBSCAN [121] and compared with each other via NMI (normalized mutual information). At a distance of 1s, clusters are barely related.
(TIF)

**S3 Fig. Uncoupled Synchrony.** When uncoupled, the global oscillator still exhibits stable clock behavior. In Upbound zones there is less phase dispersion, but no phase locking appears in uncoupled scenarios. Downbound zone oscillators act completely independently. (A) IO Spike triggered Kuramoto synchrony with average 25/75 percentiles in Downbound, after plasticity. Average STO synchrony is higher for the coupled case. STO synchrony peaks before and after IO spikes in the coupled case, but not for the uncoupled case. This indicates that gap junctions not only set a base level of synchrony, but they also induce synchronous spikes. Note the increase of synchrony before the IO spike in the coupled case. (B) Stability of the global oscillator (see S2.1B), for the uncoupled case. Mean oscillator phase has smaller variation. (C) Delta phase plot (see S2.1C) for the uncoupled case (without modulo because phase locking does not occur as in the coupled case (S2 Fig). IO neurons in the Upbound zone still keep some synchrony while for Downbound zones, oscillators behave independently.
(TIF)

**S4 Fig. Difference between coupled and uncoupled scenarios.** (A) SSpks frequency is significantly different across zones (coupled: $t = 13.9$, $p < 0.001$, uncoupled: $t = 13.61$, $p < 0.001$; two-sample Student's t-test; $n = 100$). (B) CSpks frequency (coupled: $t = 3.98$, $p < 0.001$, uncoupled: $t = 2.5$, $p = 0.014$; two-sample Student's t-test; $n = 40$). The variance of CSpks is significantly different in the uncoupled case across zones ($F = 8.65$, $p < 0.001$; F-test using CCDF of F-distribution; $n = 40$). (C) The CN firing frequency (coupled: $t = -3.26$, $p = 0.002$, uncoupled: $t = -2.98$, $p = 0.004$; two-sample Student's t-test; $n = 40$). (D) The length of the CF Pause is significantly different across zones for both coupling cases (coupled: $t = -12.52$, $p < 0.001$, uncoupled: $t = -14.39$, $p = 0.014$; two-sample Student's t-test; $n = 40$). (E) SSpk CV is significantly different for the Downbound zone across coupling cases ($t = 5.94$, $p < 0.001$; two-sample Student's t-test; $n = 100$). For the coupled case, across both zones, the SSpk CV is significantly different ($t = -4.27$, $p < 0.001$; two-sample Student's t-test; $n = 100$). (F) SSpk CV2 is significantly different for both coupled and uncoupled cas, across both zones, (Coupled: $t = -13.06$, $p < 0.001$, uncoupled: $t = -12.69$, $p < 0.001$; two-sample Student's t-test; $n = 100$). (G) CSpk CV is significantly different for the Downbound zone across coupling cases ($t = -3.69$, $p < 0.001$; two-sample Student's t-test; $n = 100$. For the coupled case, across both zones, the CSpk CV is significantly different ($t = 7.16$, $p < 0.001$; two-sample Student's t-test; $n = 100$). (H) CSpk CV2 is significantly different for the coupled case across both zones, ($t = -3.57$, $p < 0.001$; two-sample Student's t-test; $n = 100$). For the uncoupled case, the variance is significantly different across both zones ($F = 6.86$, $p < 0.001$; F-test using CCDF of F-distribution; $n = 100$).
(TIF)

**S5 Fig. Increased CN inhibition leads to reduced STO frequency and to more CSpk frequency.** (A) After Plasticity difference for SSpks is significant across coupled cases for the Upbound zone ($t = 4.51$, $p < 0.001$; two-sample Student's t-test; $n = 100$) and for coupled case across zones ($t = -5.36$, $p < 0.001$; two-sample Student's t-test; $n = 100$). The variance of SSpk frequency is significantly different for the coupled case across both zones ($F = 4.85$, $p < 0.001$; F-test using CCDF of F-distribution; $n = 100$). (B) CSpk frequency is significantly different across coupling cases (coupled: $t = 9.44$, $p < 0.001$, uncoupled: $t = 5.84$, $p < 0.001$; two-sample Student's t-test; $n = 40$) and the variance is also significantly different (coupled: $F = 2.91$, $p < 0.001$, uncoupled: $F = 12.19$, $p < 0.001$; F-test using CCDF of F-distribution; $n = 40$). (C) CN firing frequency is significantly different across coupling cases (coupled: $t = 9.15$, $p < 0.001$, uncoupled: $t = 8.87$, $p < 0.001$; two-sample Student's t-test; $n = 40$). (D) The length of the CF pause is significantly different across coupling cases for Upbound and Downbound zones (Upbound: $t = -6.25$, $p < 0.001$, Downbound: $t = 2.5$, $p = 0.013$; two-sample Student's t-test; $n = 40$). It is also significantly different across zones for the uncoupled case ($t = -7.76$, $p < 0.001$; two-sample Student's t-test; $n = 40$).

(E) Increase in CN-to-IO inhibitory synapses, and thus inhibition strength, led to a decrease in IO STO frequency and a concurrent increase in CSpk frequency.
(TIF)

**S6 Fig. Input Filter 5–10Hz.** (A) Firing frequency distributions for each population before and after plasticity for different frequency bands. (B) STO frequency vs IO spike frequency for Upbound (left) and Downbound (right) zones for before (top) and after (bottom) plasticity. There seems to be an inverse relationship between STO frequency and IO spike frequency. To reduce the effect of the spike refractory period on STO frequency, STO frequency is here defined as the mean of the 25–75 percentile of the inverse of time differences between zero crossings in the Hilbert phase. For the 5–10 Hz filter in the Upbound zone, STO frequency seems to decrease while STO spikes increase. For the Downbound zone, STO frequency is lowered after plasticity, correlated to a higher baseline CSpk frequency and thus a less pronounced difference to the 5–10 Hz filter. (C) IO spike triggered PSC for NF (top), 5–10 (middle) and 10–15 Hz (bottom) input filters. In general, IO spikes for The NF case seem to derive from entrainment followed by a final larger decrease in PSC. For 5–10 Hz filter, IO spikes seem to derive purely from entrainment to the oscillatory signal, with also a much clearer (larger amplitude) template before the spike.
(TIF)

**S7 Fig. CS-US experiment.** (A) Average weight shift after plasticity for each parallel fiber. The CS pf sees a large decrease in weight. (B) Weight changes during plasticity, averages for each CS/US trial. PF4 (CS) weight decreases continuously during CS. PFs 1,2,3 (noise) decrease during US and return quickly to baseline. PF5(US) responds the least to US. (C) Mean instantaneous Hilbert frequency of STOs, smoothed using 51ms 3rd order Savitzky–Golay filter.
(TIF)

**S8 Fig. Uncoupled CS-US experiment.** (A) Kuramoto synchrony for the uncoupled scenario (B) CS-triggered frequency density responses of the populations for SSpk (top) and CSpk (bottom). (C) SSpk frequency response after plasticity is significantly different across zones for both coupling cases (coupled: $t = -8.72$, $p < 0.001$, uncoupled: $t = -3.02$, $p = 0.003$; two-sample Student's t-test), as well as the variance (coupled: $F = 5.87$, $p < 0.001$, uncoupled: $F = 1.94$, $p < 0.001$; F-test using CCDF of F-distribution). (D) CSpk frequency after plasticity is significantly different across coupling cases in the Downbound zone ($t = -2.28$, $p = 0.026$; two-sample Student's t-test). It is also significantly different across zones for both coupling cases (coupled: $t = 7.39$, $p < 0.001$, uncoupled: $t = -5.51$, $p < 0.001$; two-sample Student's t-test), as well as the variance (coupled: $F = 1.83$, $p < 0.031$, uncoupled: $F = 10.4$, $p < 0.001$; F-test using CCDF of F-distribution).
(TIF)

**S1 Table. PC SSpk frequency statistics (n = 100).**
(DOCX)

**S2 Table. CN and IO spike frequency statistics across.** Upbound and Downbound modules
(DOCX)

## Acknowledgments

The authors would like to thank Laurens Bosman for his insights and comments on a previous version of the manuscript. We would also like to thank the NVIDIA corporation for supporting our research with the donation of 2 x RTX6000.

## Author contributions

**Conceptualization:** Elías Mateo Fernández Santoro, Chris I De Zeeuw, Aleksandra Badura, Mario Negrello.
**Funding acquisition:** Said Hamdioui, Christos Strydis, Chris I De Zeeuw, Aleksandra Badura, Mario Negrello.

**Investigation:** Elías Mateo Fernández Santoro, Chris I De Zeeuw, Aleksandra Badura, Mario Negrello.

**Methodology:** Elías Mateo Fernández Santoro, Lennart P.L. Landsmeer, Aleksandra Badura, Mario Negrello.

**Project administration:** Chris I De Zeeuw, Aleksandra Badura, Mario Negrello.

**Resources:** Chris I De Zeeuw, Aleksandra Badura, Mario Negrello.

**Supervision:** Chris I De Zeeuw, Aleksandra Badura, Mario Negrello.

**Writing – original draft:** Elías Mateo Fernández Santoro, Lennart P.L. Landsmeer, Chris I De Zeeuw, Aleksandra Badura, Mario Negrello.

**Writing – review & editing:** Elías Mateo Fernández Santoro, Lennart P.L. Landsmeer, Said Hamdioui, Christos Strydis, Chris I De Zeeuw, Aleksandra Badura, Mario Negrello.

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
