## [Decision Letter · Decision Letter 0]

22 Apr 2025

PCOMPBIOL-D-25-00184

Homeostatic bidirectional plasticity in Upbound and Downbound micromodules of the olivocerebellar system

PLOS Computational Biology

Dear Dr. Negrello,

Thank you for submitting your manuscript to PLOS Computational Biology. After careful consideration, we feel that it has merit but does not fully meet PLOS Computational Biology's publication criteria as it currently stands. Therefore, we invite you to submit a revised version of the manuscript that addresses the points raised during the review process. As reflected in the reviewers' comments, we understand that this will require a significant effort. 

Please submit your revised manuscript within 60 days Jun 22 2025 11:59PM. If you will need more time than this to complete your revisions, please reply to this message or contact the journal office at ploscompbiol@plos.org. Please include the following items when submitting your revised manuscript:

We look forward to receiving your revised manuscript.

Kind regards,

Lyle J. Graham

Section Editor

PLOS Computational Biology

**Journal Requirements:**

3) We notice that your supplementary Figures, and Tables are included in the manuscript file. Please remove them and upload them with the file type 'Supporting Information' rather than "Figure". Please ensure that each Supporting Information file has a legend listed in the manuscript after the references list.

4) Please amend your detailed Financial Disclosure statement. This is published with the article. It must therefore be completed in full sentences and contain the exact wording you wish to be published.

1) State what role the funders took in the study. If the funders had no role in your study, please state: "The funders had no role in study design, data collection and analysis, decision to publish, or preparation of the manuscript.".

**Reviewers' comments:**

Reviewer's Responses to Questions

Reviewer #1: In their study “Homeostatic bidirectional plasticity in Upbound and Downbound micromodules of the olivocerebellar system”, the authors develop a model of the olivocerebellar circuit with plasticity at the parallel fibre (PF)-Purkinje cell (PC) synapse, and analyze their model in two different parameter regimes of baseline activity corresponding to two distinct micromodule types. The authors show that in a regime with high PC and high inferior olive (IO) firing rates as postulated for downbound micromodules, PC and IO firing rates increase, respectively decrease over time with plasticity turned on, while they observe the opposite tendency in a regime with low PC and low (IO) firing rates as postulated for upbound micromodules.

Major points

While the idea of this study is interesting, I have a few difficulties with the underlying modeling logic and the presentation of the results.

- The authors motivate their study with the very general problem of stability of synaptic weights in the face of ongoing activity and continuous plasticity in recurrent networks. While it is relatively easy to understand that Hebbian plasticity in recurrent excitatory networks leads indeed to runaway activity, it is much less clear that a similar instability should exist for the olivocerebellar loop, in particular as no recurrent excitation (even if via intermediate populations) is involved.

- To address the issue of weight homeostasis, the authors study the effect of repeated presentations of relatively long (120s) bouts of frozen random input while synaptic plasticity operates, and probe activity and weights with synaptic plasticity being switched off. It is not clear how this numerical experiment, which somewhat ressembles more ‘classical’ learning scenarios with repeated inputs, relates to the problem of ‘ongoing’ plasticity in the absence of any specific learning. Do results depend on the duration of the frozen input? Do results change when the input is not frozen? Is the activity different while plasticity is ‘switched on’?

- The homeostasis is very much ‘baked in’ in the BCM plasticity rule, as the authors themselves concede. It is therefore not very surprising that synaptic weights stabilize (in addition to the first comment). The proposed BCM rule is furthermore very generic, and not constrained by experimental data on synaptic plasticity (which admittedly is limited despite a relatively huge literature on plasticity rules measured in slice, see my comment below on the cited literature). This modeling choice therefore contrasts with the declared aim of studying a biologically detailed model of the olivocerebellar system. A question would be whether the observed effects (some of which being rather small) depend on parameters of the plasticity rule?

- Getting back to the issue of ongoing plasticity and homeostasis vs. a specific learning scenario: In this work, the so-called downbound modules turn out to have lower baseline firing rates than the so-called upbound modules once plasticity has been allowed to operate. Conceptually, I would have expected these modules to be distinguishable with their respective characteristics after plasticity, or rather while plasticity operates, at least if homeostasis in the absence of any specific learning is addressed. Especially in light of this, the authors’ conclusion that the feedback loop establishes preferred activity levels for microzones (ll. 505-508) seems a bit farfetched, as simple spike rates are actually inverted.

- The authors deliberately choose to focus solely on inhibitory signaling from the CN to the IO, despite acknowledging in the discussion that the excitatory pathway may be important to consider. MF input further directly arrives at the cerebellar nuclei, which might also shape the dynamics of the micromodule feedback loop. It is unclear whether the results of the present study would hold when more elements of the circuit are taken into consideration.

General comments

- It would be helpful if the authors provided more context on the concept of upbound and downbound microzones, e.g. cite original work on individual zones and point explicitly to review introducing the concept that these zones represent distinct modules, potentially acting in concert (ll. 272-274 are unclear in this regard).

- Fig. 3A,B: The value of the comparison to experimental recordings during eyeblink conditioning is not clear (also the formulation in l. 279 is misleading as the paper doesn’t feature experimental recordings itself). First, as the authors note themselves, the timescales of the expressed changes in firing rate are incompatible. Second, in eyeblink conditioning, the changes appear over many trials, and are expressed relative to the timing of an unconditioned stimulus. It is not clear (and not described in the methods) how the changes arise in the model firing rates. Third, eyeblink conditioning is a specific learning paradigm (with specific error signals conveyed via the IO) and insofar much different from the questions the authors initially set out to address, namely weight homeostasis in the presence of ongoing plasticity.

- The references are at times unnecessarily exhaustive if not mistaken, while other parts of the literature are rather superficially touched upon:

- The first 3 refs. seem a bit off as far as they are meant as an introduction to sensorimotor plasticity.

- The “ceiling and saturation effects” referenced on l. 80 are unrelated to the specific limitations of PF-PC synaptic plasticity that has been hypothesized to support only LTD in the presence of CF signals. In contrast, the discussion of PF-PC LTD is pretty succinct if not vague (l. 82 “when PFs and CFs are activated in the same period”), referencing relatively old reviews. The picture might actually be a bit more complicated given the observed heterogeneity in PF-PC plasticity (e.g. reviewed in ref. 26), and even inverse to the classical picture of PF+CSspk = LTD, see Zang & Schutter, Curr Opin Neurobiol 2021; Bouvier et al., eLife 2019; Titley et al., J Physiol 2019.

- l 141 is an example of where 10 refs are provided for a single statement, which include a couple of modeling papers despite the statement being about experimental findings. Ref 58 furthermore seems not to be related to the referenced statement (IO phase precessions).

- Fig. 4: While in A the effects of plasticity look relatively minor, they look indeed stronger in B. This is interesting, as there are not plastic changes between CN and IO. Could the authors try to explain this observation? A related observation is that IO firing rates increase (decrease) in downbound (upbound) zones after plasticity despite an increased (decreased) activity of their inhibitory CN inputs, which is a rather counterintuitive result that the authors do not much comment on. (There is also evidence from optogenetic stimulation experiments that increased PC activity leads to increased IO discharge, Chaumont et al., PNAS 2013.) To explain their observation, in the subsequent section the authors state that “slowing down the STO creates more opportunities for spikes”. Could the authors expand on this mechanism?

Minor points

- Sentence ll. 120-128 resuming the findings is difficult to understand; better try to break into clear statements.

- l. 168 incomplete sentence

- Abbreviation PSC is not explained.

- The terminology “BCM & CSpk weights” is unclear/misleading (they describe weight changes if I understood the model correctly).

- Are the hard weight constraints reached? If yes, by what fraction of weights?

- Fig. 2C: IO spiking seems pretty bursty - did the authors characterize this burstiness and can compare to experimental data?

- How much do results depend on synaptic delays? They can be expected to vary, and are not known with ms precision (much shorter times have been reported for PC-CN connections, ref. 43).

- How much do results depend on BCM rule parameters? Did the authors perform some kind of sensitivity analysis?

- Fig. 1A is unclear / very difficult to understand.

- The notion of “oscillating” weights is unclear. Do these weight oscillations depend on the length of the stimulus w/ plasticity (120s)? Is there a stable weight distribution for ongoing stimulation (w/o repeats of frozen noise)? (Those oscillations look small relative to the absolute weights btw.)

- Fig. 4: title: It seems the resonances are zone-dependent also before plasticity; slightly misleading graphics as Cspk-triggered data are aligned but have different x-axis.

- ll. 357+ suggests that the effects reported in Fig. 4 are not caused by the entire feedback loop, is this true?

- ll 370/71: What dos frequency-band dependent variation with frequency mean?

- Fig. 6: “phase period” unclear. “zebrin positive/negative” → better use previous terminology of down/upbound zones. “NF” → provide “no filter” in legend for better readability.

- ll 577-578: It is surprising that variability in IO spike rates disappears in the absence of gap junction; Fig. 7 furthermore seems to suggest the contrary. Am I missing something?

- l. 461: To my understanding, the authors studied a model in two different parameter regimes, that are meant to correspond to upbound and downbound micromodules. In the absence of any interaction or integration of both in a larger network, the formulation of a cerebellar model comprising both modules is misleading.

- The title should reflect that this study investigates a model and doesn’t report experimental findings on the cerebellum (e.g. “A model of homeostatic bidirectional plasticity in Upbound and Downbound micromodules of the olivocerebellar system”).

- The methods are partly sloppily written. Please fix:

- l. 602: Gaussian white noise

- Table 1: units of \sigma_OU

- l. 622: Sampling of parameter values is uclear: uniform? evenly spaced?

- Eq. numbers are shifted / don’t correspond

- l. 644: firing → current

- Some formulations misleading: l. 645 “current is 0 nA when there is no spike” although it is a filtered version of the spike train and will hardly be 0 ever.

- the nomenclature changes at some places (confusing subsubscripts)

- l. 668: The reference to the eyeblink experiment is unexpected - if description of simulation underlying Fig. 3 please be much more explicit on how those were done.

- Tables 2, 4: Several parameter variations are very small and can be expected to be irrelevant, while others are much more important, whereas no variations at all are considered for the values given in Table 6. Please comment.

- Presentation of BCM rule sloppy (what is “our BCM”?); the main equations are not the ones used, dimensions are inconsistent (Eq. 16,17) (have only currents for PF a priori, no def of \rho_{PF}, \rho_{PC} given) ; l 739 minimum wrong.

- l. 760: “instantaneously” contradicts long synaptic delay

- l. 772: release of GABA into the cell?!

- Eq 26: V measured in which units?

- ll 798/99 unclear

- More generally, the two-step presentation of the model is somehow misleading, with first a definition of a baseline model (Table 6) and then different parameter values given in Table 7.

- l 827/28 unclear: What was done to the weights, and why?

Reviewer #2: This paper delivers a computational model to explore how synaptic plasticity dynamics is stabilized during cerebellar learning. The model focuses on two types of cerebellar micromodules: Upbound and Downbound. These modules differ in firing rates and plasticity tendencies: Upbound Purkinje cells (PC) tend to potentiate synaptic strength, while Downbound PCs tend to depress synaptic strength. The model thus shows that synaptic weights stabilize over time, and that IO oscillations and coupling influence the frequency ranges at which parallel-fiber inputs can be optimally stabilized. The model suggests that inferior olive driven temporal dynamics help coordinate plasticity across cerebellar modules.

The paper provides a tool to test different hypotheses of interest for those researchers working on cerebellar function and learning. I find several results of this paper relevant for publication. However I also find several weaknesses:

1-This is a rather lengthily paper hard to read because of its organization. A minimal explanation of the model is required to understand its assumptions, the provided synaptic plasticity exploration and the results. However key model descriptions come at the very end of the paper, and thus readers unfamiliar with the authors' work will be completely lost. The most notorious example of this is the use of the term "synaptic weight" in this model, which is first explained on page 40, but it is used from the very beginning.

2-The paper uses very simple adaptive exponential integrate and fire models to describe PCs and cerebellar nuclei (CN) and a three-compartment biophysical description of the IO cells. An explanation is required why PC morphology is not an issue for the conclusions reached in this model, as well as the simple dynamics for all synapses beyond the plasticity models used.

3-Stochastic input should be better justified for the purpose of the plasticity exploration, particularly in a system where functional input timing and correlations can be essential for the learning process.

4-In general, it is not clear what results required fine tuning in the model, and what results are robust for a wide enough parameter space. Hypotheses are drawn but then the model seems to be tuned to lead to the phenomena hypothesized.

5-Overall, limitations of the modeling approach beyond what the model left out should be discussed in the view of the conclusions.

Minor issues and suggestions to improve readability and impact:

6-Please justify the chosen ratio of PCs, CN and IO cells.

7-Fig. 2: please specify units for the synaptic weights.

8-Fig. 3A: no vertical scale is shown for the experimental recordings. Please explain the difference in timescale with the model results shown in B.

9-Please justify the different statistical tests used to present the model results thought out the paper.

10-For the reader it is not clear where the PSC is being measured in Fig. 4 (by the way the acronym is not defined until much later). In general in Fig. 4 and throughout the paper, it is not clear what experimental results require fine tuning to be reproduced.

11-Line 644: Eq5 -> Eq 6.

12-Line 820: Sampling rate is not an integration timestep. Please clarify this issue.

13-Please indicate how many spikes were used for the analysis shown in Fig. S3.

14-Please add the code GitHub link in the text.

**Have the authors made all data and (if applicable) computational code underlying the findings in their manuscript fully available?**

Reviewer #1: None

Reviewer #2: Yes

PLOS authors have the option to publish the peer review history of their article (what does this mean?). If published, this will include your full peer review and any attached files.

Reviewer #1: No

Reviewer #2: No

**Figure resubmission:**

**Reproducibility:**



---

## [Decision Letter · Decision Letter 1]

27 Jul 2025

Homeostatic bidirectional plasticity in Upbound and Downbound micromodules in a model of the olivocerebellar loop

PLOS Computational Biology

Dear Dr. Negrello,

Thank you for submitting your manuscript to PLOS Computational Biology. After careful consideration, we feel that it has merit but does not fully meet PLOS Computational Biology's publication criteria as it currently stands. Therefore, we invite you to submit a revised version of the manuscript that addresses the points raised during the review process, in particular addressing the concerns of Reviewer 1.

Please submit your revised manuscript within 60 days Sep 26 2025 11:59PM. If you will need more time than this to complete your revisions, please reply to this message or contact the journal office at ploscompbiol@plos.org. Please include the following items when submitting your revised manuscript:

We look forward to receiving your revised manuscript.

Kind regards,

Lyle Graham

Section Editor

PLOS Computational Biology

**Reviewers' comments:**

Reviewer's Responses to Questions

Reviewer #1: While the overall clarity of the manuscript has indeed been improved, my main questions/difficulties with the study have not been satisfactorily addressed.

Main points

1. “Baseline homeostasis” of up- and downbound modules in the presence of ongoing activity vs. procedural learning - the manuscript remains ambiguous which of these two effects are actually addressed, and remains short of an answer of how both experimental observations might be reconciled.

2. How does increased CN input increases IO output / CF rate? The authors propose ‘likely’ causes without providing evidence in that sense, and refer to experimental literature that has not shown such an effect. On the contrary, the optogenetic study by Chaumont et al. suggests that decreased CN activity (by stimulation of upstream PCs) leads to increased IO discharge.

3. The authors do not test how their model responds to repeated exposures to new instantiations of OU input. A comparison between such a scenario and theirs remains crucial to assess what part of the observed stabilisation is due to the repeated temporal contingencies in the input (eventually relatable to learning) and what part is due to continuously fluctuating inputs (addressing the issue of weight homeostasis).

4. The authors report in their reply that “expected biological behavior” of firing rate changes is only observed for a “narrow range” of threshold timescale values. This suggests the model has been fine-tuned at least to some degree. The range of values should be reported, the value related to known plasticity mechanisms and the discussion not hidden in the methods (btw. on l. 908-917 and not where stated in the reply).

5. The model description is somewhat improved, but several points raised in the first review have not or only insufficiently been addressed.

Comments to the updated MS

- l. 85: “In this model” refers to which model? Connection not clear.

- l. 87-88, l. 109-119: Resumes perfectly well the issue I have with the concept of homeostasis presented here. The long-term adaptation to input statistics should be the background on which actual learning operates. The feedback loop is studied without any coupling to motor errors or any other output-related, environment-mediated CF feedback; only intrinsic activities are considered. Under the hypothesis that plasticity is continuously operating and not switched on on a case-by-case basis, the feedback loop should indeed set a stable, “homeostatic” baseline state. In this setting, I would expect upbound and downbound modules to be the resulting baseline states as they are observed experimentally, not simply initial conditions before plasticity operates. Whether those resulting baseline states then further behave as observed experimentally during learning (with some form of learning-related, specific CF feedback), is another question. A true insight would be how these two experimental observations (stable, homeostatic modules, with corresponding biased firing rate modulations upon learning) can be reconciled.

- l. 120-146: Somewhat in contradiction to what preceeds, the authors relate their model here more directly to procedural learning, suggesting that they study the actual process of learning rather than homeostasis during ongoing activity. This apparent inconsistency needs to be clearly resolved. More generally, it seems that the authors do not (potentially cannot) address how both types of plastic changes (ongoing & homeostatic, leading to stable baseline states on the one hand, and learning-related, leading to modified weight distributions and firing rates on the other hand) can coexist.

- l. 165-167: “test repeated exposures to the same input” and especially l. 171-171 “a crucial aspect of our model is that we repeated presentations” - In its current form, the article does not allow the reader to appreciate whether the results depend on stimulus repetition or not; this should be clearly stated and shown.

- Fig. 2A: It is still not clear how to read this plot. What does the y-axis represent? What is represented by the gray-value shading? The color map doesn’t have a label, please also provide a unit for numerical values.

- Fig. 2D-F: Weight or weight update (in which units)? It is not clear why “CSpk weights” (what does this mean?) should decay to baseline. Also, please replace “theta” by the appropriate greek letter throughout the figure caption (as used in the MS and the figure itself).

- l. 317: Not clear from Fig. S2-1A how the global phase and “delta” phase are determined.

- l. 350: “these results” is not clear. What results is this referring to?

- Fig. 3A,B: This comparison still seems meaningless/misleading to me. The authors now seem to suggest so themselves in the main text (l. 339-349), which makes it even harder to understand why they kept this figure. If I would try to follow their (possible) reasoning and make sense of this comparison, then perhaps the timecourse of the SSpk rates AFTER stimulus onset over the course of learning, as indicated by the white arrows, might correspond to the (slow) changes in the SSpk firing rate during plasticity. This cannot be appreciated by this plot (besides the more general issue of learning vs. spontaneous weight updates).

- Fig. 3B: What kind of averaging or smoothing do the thick lines represent in the model part? It is unclear how the firing rates can decrease before the onset of plasticity.

- l. 372: “indemnified” → identified

- Fig. 3C-E: It is not clear how model-cells of both modules (parameter sets) relate to each other, as suggested by the gray lines (no pre-post paradigm, treatment conditions etc.). Please explain or remove lines.

- l. 390: table states 50 ms CN-IO synaptic delay vs. 60 ms here

- l. 398-399: Not clear how Fig. S3E supports the statement.

- l. 399-401: Repetition, and no evidence given.

- l. 402-403: “how amplitude of CN-triggered IO responses were more sensitive to PF-PC plasticity.” unclear. More sensitive than what?

- l. 403: “This variability in CSpk timing” → variability of CSpk timing hasn’t been mentioned before, so unclear what this refers to.

- l. 425: “CN inhibition” potentially misleading (I misread as “inhibition of CN”), replace by “inhibitory CN input”?

- l. 450-451: Which elements of the circuit (feedback loop) have been disregarded before?

- l. 498-499: Still not clear. If one assumes that the “spike opportunity” for a STO corresponds to the time spent above a given voltage threshold, then for a decreased frequency obviously the time above that threshold increases, but so does the time below threshold. The relative proportion of time, which to first order corresponds to the average excitability or “spiking opportunity” will remain the same.

- l. 517,518: Please replace terminology “zebrin-positive/negative” in line with response to comment in round 1.

- l. 550, 553, 586, probably other instances: “continual” → continuous

- l. 560-561: With the formulation “intrinsic learning”, it is again not clear whether learning or homeostasis in the presence of ongoing plasticity is addressed.

- l. 585: “weights that oscillated” is still not clear. Oscillations imply that there exists a period of that oscillation. Please characterize the strength of the weight fluctuations across epochs (ideally as a function of epoch length, and for repeated/frozen vs. newly drawn patterns, see also previous comment to l. 165-167).

- l. 595: “underdamped” suggests indeed an underlying oscillatory dynamical system, which is not really shown here, see again earlier comments.

- The methods have improved, but the model description remains sloppy, e.g. l. 765-766: How can it be that superscript i = PC, CN when the sum is carried out over i and cell-type specific currents?

- l. 819: “differential equation {I_{IO}}_{CN}” What does this refer to? Also, please make sure to keep notations consistent: I^{(OU)}_{XY} is not the same as {I_{OU}}_{XY} (superscript with parenthesis vs. subscript without).

- l. 876: symbol missing/wrongly represented

- l. 893: Why is \phi indexed by i?

- l. 888: What is the instantaneous firing rate of the PF? Their activity has been described as an OU process representing current input, not fluctuating firing rates.

- l. 898: “We need to damp this” - why? Formulation sounds funny for a methods section. Please make very explicit which parts of the BCM rule have been introduced earlier (relate to the original model or variants) and which parts / modeling choices are introduced by the authors.

- Eq. 18: Doesn’t make sense, as the argument of tanh has the dimension of Hz. Please correct.

- l. 904: Comparison with 0 doesn’t make sense as written (should be difference).

- Eq. 20: Dimension mismatch between \rho_{PF} and I_{PF}. Also, why is the change relative to deviation from mean PF input, with equally strong weight changes for stronger and lower PC activity?

- More generally regarding the LTD weight changes: If w_CSpk represents a weight and not a weight change (question asked in first review, see also points above), then this rule also incorporates (aspects of) LTP because of the decay of the w_CSpk component (which correspond to positive weight changes). Please give a very decent thought to the definition and description of the plasticity rules.

- l. 999-1006: Why was the temporal filtering necessary, when only instantaneous snapshots of weights are compared? How would the results change when no filtering was used (and unfiltered weights used as inputs for the next epoch)?

Reviewer #2: The authors have addressed all my concerns raised in the first review iteration. The revised manuscript is significantly improved and provides clearer explanations of the model assumptions, derived results, and limitations. Units are missing in table 7. Otherwise, the paper is now ready for publication. I enjoyed reading this latest version of the manuscript. Congratulations on your work.

**Have the authors made all data and (if applicable) computational code underlying the findings in their manuscript fully available?**

Reviewer #1: None

Reviewer #2: Yes

PLOS authors have the option to publish the peer review history of their article (what does this mean?). If published, this will include your full peer review and any attached files.

Reviewer #1: No

Reviewer #2: No

**Figure resubmission:**

**Reproducibility:**



---

## [Decision Letter · Decision Letter 2]

2 Sep 2025

PCOMPBIOL-D-25-00184R2

Homeostatic bidirectional plasticity in Upbound and Downbound micromodules in a model of the olivocerebellar loop

PLOS Computational Biology

Dear Dr. Negrello,

Thank you for submitting your manuscript to PLOS Computational Biology. After careful consideration, we feel that it has merit but does not fully meet PLOS Computational Biology's publication criteria as it currently stands. Therefore, we invite you to submit a revised version of the manuscript that addresses the points raised during the review process.

Please submit your revised manuscript within 30 days Nov 02 2025 11:59PM. If you will need more time than this to complete your revisions, please reply to this message or contact the journal office at ploscompbiol@plos.org. Please include the following items when submitting your revised manuscript:

We look forward to receiving your revised manuscript.

Kind regards,

Lyle J. Graham

Section Editor

PLOS Computational Biology

**Reviewers' comments:**

Reviewer's Responses to Questions

Reviewer #1: I acknowledge the effort the authors put into trying to address my concerns. Although the manuscript still suffers from some ambiguity regarding homeostasis vs. learning pointed out in my previous reports, I do now consider the manuscript in principle fit for publication as important clarifications have been made.

Just in addition to my earlier reports, I would like to suggest the following thought experiment: Stick an electrode in a characteristic downbound or upbound zone and measure SSpk or CSpk rates in awake animals in the absence of any learning experiment being performed. What would be the measured baseline firing rates? Would these rates evolve over time? Would the authors argue that plasticity is operating or not during and in between recording sessions?

If the authors agree that firing rates remain stable and that plasticity is not gated during learning but operating continuously, this is the baseline regime that the authors should describe with their model *after* ongoing plasticity has lead to stationary weight distributions, and based on which they should distinguish up- and downbound modules according to their physiological criteria. In contrast, any hypothetical “initial”, before-plasticity network state should be considered experimentally inaccessible.

It may well be that I continue to miss something in the authors’ logic - but taken at face value, the final stationary SSpk rates of both modules that the authors report are actually inverted relative to experimental observations, i.e., the authors report higher baseline SSpk rates for the upbound module and lower baseline SSpk rates for the downbound module (Fig. 4C). In light of this result, the statement on l. 711 (“loop mechanisms enhance this functional baseline difference”) appears unwarranted if not wrong. Furthermore, it remains unclear, and I am obviously repeating myself, why the initial relaxation towards these baseline states is deemed comparable to any learning-related firing rate changes (as in Fig. 3A,B, notwithstanding the more clearly stated caveats), or more generally interpreted as plasticity-mediated stabilization of the modules’ baselines. The comparisons of Fig. 3, which are suggestive of a “match” between experiment and model, are misleading in that regard; to avoid any confusion, it should at least be stated in the legend of Fig. 3C-E that the model firing rates do not correspond to the homeostatic baseline state (after plasticity) but to the initial rates before plasticity is allowed to operate.

I appreciate that the authors now included additional simulations of actual learning on top of the baseline states obtained by ongoing plasticity. One may argue whether the implemented protocol, in which e.g. the strength of the US relayed to the IO does not depend on the PC activity, is a faithful implementation of cerebellar learning or eyeblink conditioning in particular, as increased eyelid closure should result in lesser stimulation of the IO. (Relatedly, it is not clear whether the authors considered this interaction in their comment ll. 612-614.) However, I understand that a more detailed simulation of eyeblink conditioning goes beyond the scope of this article. The data of Fig. 7F seem to suggest that the SSpk rates in up- and downbound zones do indeed change as expected during learning, which is a nice result. It would be helpful if the authors could state more clearly in the legend or methods how and when those rates were determined, as this cannot be deduced from Fig. 7E (where the effect is not clearly visible btw.).

One last point: On ll. 1014-1015 the authors now state that the effective PF firing rate is computed from the amplitude of the OU-driven synaptic current, but exactly how this calculation is done is not specified.

**Have the authors made all data and (if applicable) computational code underlying the findings in their manuscript fully available?**

Reviewer #1: None

PLOS authors have the option to publish the peer review history of their article (what does this mean?). If published, this will include your full peer review and any attached files.

Reviewer #1: No

**Figure resubmission:**
---

## [Decision Letter · Decision Letter 3]

7 Oct 2025

PCOMPBIOL-D-25-00184R3

Homeostatic bidirectional plasticity in Upbound and Downbound micromodules in a model of the olivocerebellar loop

PLOS Computational Biology

Dear Dr. Negrello,

Thank you for submitting your manuscript to PLOS Computational Biology. After careful consideration, we feel that it has merit but does not fully meet PLOS Computational Biology's publication criteria as it currently stands. Therefore, we invite you to submit a revised version of the manuscript that addresses the points raised during the review process.

Please submit your revised manuscript within 30 days Dec 07 2025 11:59PM. If you will need more time than this to complete your revisions, please reply to this message or contact the journal office at ploscompbiol@plos.org. Please include the following items when submitting your revised manuscript:

We look forward to receiving your revised manuscript.

Kind regards,

Lyle J. Graham

Section Editor

PLOS Computational Biology

**Reviewers' comments:**

Reviewer's Responses to Questions

**Comments to the Authors:**

Reviewer #1: Please remove the sentence “loop mechanisms enhance this functional baseline difference”, which is not warranted by the simulation results, see my earlier reports. In their last reply, the authors indeed acknowledge that their results differ with experimental data in magnitude and direction (direction being the important part here, as the authors claim that the baseline difference is *enhanced* when their simulations show the opposite). In their reply, the authors seem to diminish the importance of this difference by referring to “some” experimental data, but the entire argument of their study is actually built on an observed baseline difference with strongly discharging downbound modules and lesser discharging upbound modules (see e.g. ll. 94-108).

In ll. 600-601, the authors state that they will investigate how the previously identified baseline states evolve during a simulated learning paradigm. I apologize for not having paid more attention earlier, but it is not clear then how individual and mean firing rates of both modules as shown in Fig. 7E before CS onset and Fig. 7F left can be inverted relative to the homeostatic baseline firing rates reported in Figs. 4C & 5B (no filter). In Fig. 7, the baseline difference in density is ~0.45 (which units?) for the downbound module vs. ~0.3 for the upbound module, which amounts to a significant difference in baseline mean firing rates of approx. ~1.5 downbound/upbound, in marked contrast to roughly equal firing rates at baseline reported earlier (see also ll. 471-473).

**Have the authors made all data and (if applicable) computational code underlying the findings in their manuscript fully available?**

Reviewer #1: None

PLOS authors have the option to publish the peer review history of their article (what does this mean?). If published, this will include your full peer review and any attached files.

Reviewer #1: No

**Figure resubmission:**
---

## [Editor Report · Decision Letter 4]

12 Oct 2025

Dear Dr. Negrello,

We are pleased to inform you that your manuscript 'Homeostatic bidirectional plasticity in Upbound and Downbound micromodules in a model of the olivocerebellar loop' has been provisionally accepted for publication in PLOS Computational Biology.

Best regards,

Lyle J. Graham

Section Editor

PLOS Computational Biology

---

## [Editor Report · Acceptance letter]

PCOMPBIOL-D-25-00184R4

Homeostatic bidirectional plasticity in Upbound and Downbound micromodules in a model of the olivocerebellar loop

Dear Dr Negrello,

I am pleased to inform you that your manuscript has been formally accepted for publication in PLOS Computational Biology. Your manuscript is now with our production department and you will be notified of the publication date in due course.

With kind regards,

Anita Estes
